# ICaRus: Identical Cache Reuse for Efficient Multi Model Inference

**Sunghyeon Woo**[*]**, Jaeeun Kil**[*]**, Hoseung Kim, Minsub Kim, Joonghoon Kim,**
**Ahreum Seo, Sungjae Lee, Minjung Jo, Jiwon Ryu, Baeseong Park, Se Jung Kwon,**
**Dongsoo Lee**

NAVER Cloud
{sunghyeon.woo1, jaeeun.kil}@navercorp.com

## Abstract

Multi model inference, where multiple task-specialized models collaborate to solve complex real-world problems, has recently emerged as a prominent paradigm, particularly in the development of agentic AI systems. However, in such scenarios, each model must maintain its own Key-Value (KV) cache for the identical prompt, leading to substantial memory consumption. This explosive growth of KV caches forces LLM serving systems to evict previously stored caches, which in turn introduces significant recomputation overhead whenever the evicted caches are required again. Moreover, prefix caching is inherently infeasible across different models, forcing each model to recompute KV cache for the identical prompt, which leads to significant overhead. To alleviate these issues, we propose **I**dentical **Ca**che **Reus**e (**ICaRus**), a novel architecture that allows multiple models to share identical KV caches across all layers. ICaRus is based on the key observation that a decoder-only Transformer can be conceptually decomposed into a logical encoder, which generates KV caches, and a logical decoder, which predicts output tokens from the KV caches. ICaRus fine-tunes only the logical decoder while freezing the logical encoder, enabling multiple models to share an identical KV cache. This eliminates cache memory explosion and unexpected evictions while also allowing cross-model reuse of KV caches for new input tokens, thereby removing redundant recomputation in multi model inference achieving both efficiency and scalability. Moreover, by incorporating lightweight adapters such as LoRA, ICaRus parallelizes KV cache generation and next-token prediction during decoding. ICaRus achieves comparable accuracy to task-specific fine-tuned model across a diverse set of tasks, while allowing multiple specialized models to fully share KV caches. ICaRus achieves up to $11.1\times$ lower P95 latency and $3.8\times$ higher throughput in multi agent workflow with 8 different models, compared to conventional multi model system.

## 1 Introduction

Large Language Models (LLMs) have shown strong performance across domains (Zhao et al., 2024; Dubey et al., 2024; Comanici et al., 2025; Yang et al., 2025); however, a single model struggles with complex tasks that demand multi step reasoning and domain-specific expertise (Tang et al., 2020; Yao et al., 2023; Sun et al., 2024). Recently, the emerging paradigm of multi model inference addresses this limitation by orchestrating task-specialized models, achieving higher accuracy and problem-solving ability than a general-purpose model (Fu et al., 2023; Du et al., 2024; Shen et al., 2024; Subramaniam et al., 2025). However, this paradigm introduces severe challenges in managing the Key-Value (KV) cache: each model maintains its own cache even for identical prefixes, causing memory consumption to grow rapidly with the number of models. Once GPU memory is saturated by KV cache, serving systems (Kwon et al., 2023; Zheng et al., 2024) must evict caches, which triggers redundant recomputation and significantly degrades throughput. Furthermore, because KV caches are model-specific, prefix caching (Kwon et al., 2023; Zheng et al., 2024) cannot be applied

---
[*]Equal contribution

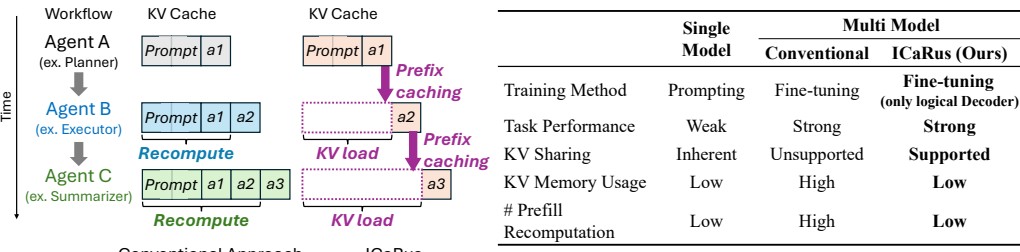

(a) KV Cache management strategies in agent workflow using multi model

(b) Comparison of ICaRus and conventional approaches

Figure 1: Comparison of KV cache management strategies and effectiveness in multi model scenarios between conventional approaches and ICaRus.

across different models, which forces identical prompts to rebuild KV caches independently and thereby increases latency.

Previous KV cache optimization techniques, such as pruning (Zhang et al., 2023), quantization (Hooper et al., 2024; Yang et al., 2024), and inter-layer sharing (Qiao et al., 2024), reduce cache size while minimizing accuracy degradation. Unlike traditional LRU-based prefix caching, KVFlow (Pan et al., 2025) schedules KV cache eviction and prefetching based on anticipated agent workflow, reducing recomputation overhead. However, these methods focus only on single model cache management, leaving unresolved the challenges of cache explosion and the lack of KV cache sharing of prefix in multi model settings. DroidSpeak (Liu et al., 2024b) addressed multi model KV cache management by sharing non-sensitive layer caches between a base model and its fine-tuned variants, thereby reducing recomputation cost. However, this approach has inherent limitations, as caches from sensitive layers remain unshared and must still be recomputed.

To address these issues, we propose **I**dentical **Ca**che **R**euse (**ICaRus**), a novel architecture that enables multiple models to share and reuse the same KV cache across all layers. The core idea of ICaRus originates from conceptually decomposing a decoder-only Transformer into two parts: a logical encoder, which is responsible for generating KV cache, and a logical decoder, which predicts the next token from the cache. We freeze the logical encoder of pretrained LLM (i.e. base model) and fine-tune only the logical decoder. Since all specialized models share the identical logical encoder, the KV cache generated for an identical prompt is likewise identical, enabling direct sharing without redundant memory usage as shown in Fig. 1(a). This prevents GPU memory from rapidly saturating due to KV cache growth, avoiding costly recomputation caused by cache eviction. Moreover, shared KV caches enable prefix caching across models, eliminating redundant computation for identical prompts and further improving efficiency as depicted in Fig. 1(b). In addition, ICaRus leverages the adapter architecture to generate the KV cache for the next step in parallel with the next-token computation during the decode phase. We evaluate ICaRus across diverse tasks including mathematics, coding, and knowledge understanding on a wide range of model families and scales (LLaMA-3.1-8B, Qwen3-1.7B/8B/14B). The results demonstrate that ICaRus achieves accuracy comparable to task-specific fine-tuned models, even though ICaRus-tuned models are able to share KV caches across tasks. Furthermore, when integrated into the vLLM serving system and evaluated in multi agent scenario, ICaRus delivers as much as a $11.1\times$ reduction in 95th-percentile (P95) latency and a $3.8\times$ throughput gain compared to conventional multi model system.

In summary, the main contributions of this work are as follows:

- We propose ICaRus, the first architecture that enables multiple decoder-only Transformers to fully share KV caches, guaranteeing high generation quality in real serving scenarios by explicitly modeling the fully shared-KV setting already at training time.

- We demonstrate that ICaRus achieves accuracy comparable to task-specific fine-tuning across diverse tasks (mathematics, coding, and knowledge understanding) and model architectures (LLaMA-3.1-8B, Qwen3-1.7B/8B/14B).

- We confirm that ICaRus significantly improves efficiency in multi agent workflow, achieving up to $11.1\times$ reduction in P95 latency and $3.8\times$ improvement in throughput compared to conventional multi model system.

## 2 BACKGROUND & MOTIVATION

**Key-Value Cache in LLM Serving Systems.** During autoregressive inference, decoder-only Transformers generate tokens sequentially, where each new token depends on all previously generated tokens. Computing self-attention naïvely for every step requires recomputation over the entire sequence, incurring a per-token complexity of $\mathcal{O}(n^2)$ where $n$ is the sequence length. To avoid this quadratic overhead, modern LLM serving systems cache the key and value representations of previously processed tokens (Vaswani et al., 2017). By reusing these cached states, each new decoding step only attends to the most recent token, reducing the per-token attention complexity to $\mathcal{O}(n)$ and thereby significantly lowering computational cost. However, the size of KV caches grows linearly with both sequence length and model depth, imposing substantial memory pressure on GPU-based serving systems (Kwon et al., 2023; Zheng et al., 2024). Consequently, memory-efficient cache management has emerged as a critical challenge for scalable LLM deployment.

**Prefix Caching in LLM Serving Systems.** Prefix caching is a widely adopted optimization that reuses the KV cache corresponding to a fixed prefix across multiple queries sharing the same initial context (Kwon et al., 2023; Zheng et al., 2024). This technique is particularly effective in scenarios such as retrieval-augmented generation (RAG) (Lewis et al., 2020) and instruction-tuned applications (Chung et al., 2024; Ouyang et al., 2022), where prompts often contain long but invariant components like system prompts, task-specific templates, or retrieved documents. By reusing the cached key-value states of these repeated prefixes, serving systems can avoid redundant computation during the prefill phase, effectively reducing the computational complexity from $\mathcal{O}(n^2)$ to $\mathcal{O}(mn)$, where $n$ denotes the sequence length and $m$ denotes the variable suffix length with $m \ll n$, thereby improving both throughput and latency. Moreover, prefix caching is highly beneficial in multi-turn conversational settings, where a large dialogue history is preserved across turns and only the most recent user utterance changes; by caching the KV states of the shared history and computing attention only for newly appended tokens, serving systems can efficiently support interactive dialogues without recomputing the entire context at every turn (Kim et al., 2025).

**Agentic AI Workflow and Multi Model Inference.** Agentic AI and workflow-based reasoning have given rise to complex pipelines in which models are orchestrated to perform specialized roles. For instance, ReAct (Yao et al., 2023) alternates between *Thought → Act → Observation*, Reflexion (Shinn et al., 2023) incorporates self-evaluation loops, LATS (Zhou et al., 2024) explores reasoning through parallel branch expansion, and LLMCompiler (Kim et al., 2024) constructs a DAG to schedule overlapping tool and model calls. When executed within a single model, such workflows can leverage prefix caching to avoid redundant computation, thereby reducing effective memory usage, lowering P95 latency, and improving throughput (Kim et al., 2025). However, in multi model settings where task-specialized models collaborate within a single pipeline, each model must maintain its own KV cache even for identical prefixes. Such KV cache duplication leads to memory usage that grows linearly with the number of active models; once GPU capacity is saturated, this growth inevitably triggers cache eviction, which in turn forces recomputation of evicted prefixes. Moreover, since prefix caching typically operates only within individual models, identical prefixes must be recomputed separately across models, leading to redundant prefill computation that inflates both latency and energy consumption. These limitations underscore the need for new architectures that support cross-model KV sharing and prefill de-duplication in multi model inference.

## 3 DESIGN OF ICARUS

### 3.1 DECODER-ONLY TRANSFORMER AS LOGICAL ENCODER AND DECODER

We first present a mathematical formulation of the decoder-only Transformer, which predicts the next token conditioned on the current token context. Specifically, we abstract $x_i$, $k_i$, and $v_i$ as the $i$-th token, its key representation, and its value representation, respectively, and denote the decoder-

only Transformer by $F$. In this case, the next-token generation from the current token context in a decoder-only Transformer can be expressed as $x_{i+1} = F(x_1, x_2, \ldots, x_i)$. To generate the next token $x_{i+1}$, the model requires two types of information: the current token $x_i$ and the accumulated key–value pairs. We denote the key set and value set up to step $i$ as $K_{1:i} = \{k_1, k_2, \ldots, k_i\}, V_{1:i} = \{v_1, v_2, \ldots, v_i\}$. More concretely, in the attention operation, the query derived from $x_i$ is generated anew at each step, whereas the keys and values are continuously appended to the cache and reused across subsequent decoding steps. In other words, the query does not persist beyond its step, but the KV pairs accumulate and form the long-term memory. This dependency can be expressed as

$$x_{i+1} = F(x_1, x_2, \ldots, x_i) = F(x_i, \ K_{1:i}, V_{1:i}). \tag{1}$$

Eq.1 indicates that a decoder-only Transformer predicts the next token conditioned on the current token $x_i$ and the KV cache constructed up to this point. More generally, the generation process can be decomposed into two conceptual stages: (1) constructing the key set $K_i$ and the value set $V_i$ from the input sequence $x_{1:i} = \{x_1, x_2 \ldots, x_i\}$, and (2) decoding the next token $x_{i+1}$ based on the current token $x_i$ together with the accumulated sets $(K_i, V_i)$. Formally, this can be expressed as

$$K_{1:i}, V_{1:i} = E(x_{1:i}), \tag{2}$$
$$x_{i+1} = D(x_i, \ K_{1:i}, V_{1:i}), \tag{3}$$

where we introduce a logical encoder, denoted by $E$, that transforms the input sequence into its key and value representations, thereby constructing the KV cache, and a logical decoder, denoted by $D$, that consumes the current token and the KV set to generate the next token. Importantly, a decoder-only Transformer can be interpreted as the special case where the parameters of the logical encoder and logical decoder are identical. More detailed concept of logical encoder-decoder architecture is depicted in Appendix C.

## 3.2 ICaRus: Identical Cache Reuse across LLMs

As described in Section 3.1, a decoder-only model can be decomposed into a logical encoder, which generates key–value pairs from a given token, and a logical decoder, which predicts the next token using the current token and the accumulated KV cache, as shown in Eqs. 2–3. From this perspective, task-specific fine-tuning can be viewed as jointly training both the logical encoder and the logical decoder to specialize in a given task. While such task-tuned models achieve strong task-specific capabilities, each maintains its own logical encoder thereby preventing KV cache sharing even when prompts are identical across models.

Building on this insight, we propose the ICaRus architecture which fine-tunes only the logical decoder of a decoder-only Transformer as below.

$$K_{1:i}, V_{1:i} = E_{base}(x_{1:i}), \tag{4}$$
$$x_{i+1} = D_{task}(x_i, \ K_{1:i}, V_{1:i}), \tag{5}$$

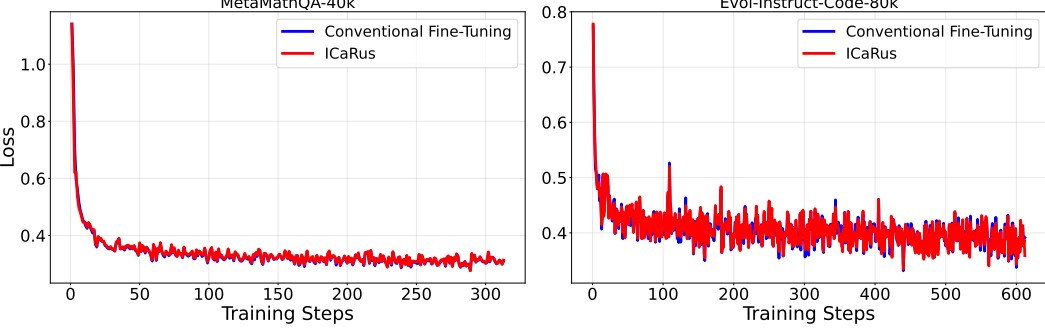

Figure 2: Training loss curves of conventional fine-tuning and ICaRus, both applied with LoRA on LLaMA-3.1-8B, trained on the MetaMathQA-40k and Evol-Instruct-Code-80k dataset.

Here, $E_{base}$ is the frozen logical encoder inherited from the base model, and $D_{task}$ is the logical decoder fine-tuned for the target task. Specifically, the logical encoder and the logical decoder are initialized with the parameters of the base model, a pretrained decoder-only Transformer (i.e. $F_{base} \equiv E_{base} \equiv D_{base}$). The task-specific logical decoder $D_{task}$ in Eq. 5 is then trained, starting from the base decoder $D_{base}$, to predict the next token $x_{i+1}$ under two objectives: (1) specializing in the target task, and (2) leveraging the KV cache generated by the frozen logical encoder in Eq. 4. As a result, multiple task-specific logical decoders (e.g., $D_{\text{math}}$, $D_{\text{coding}}$, $D_{\text{reasoning}}$) can share a single logical encoder, which is identical to the base model (i.e., $E_{\text{math}} \equiv E_{\text{coding}} \equiv E_{\text{reasoning}} \equiv E_{base}$), thereby enabling all models to reuse the identical KV cache generated by the shared encoder, as illustrated in Fig. 1.

During training, the input data are duplicated and provided to both the logical encoder and the logical decoder. The logical encoder generates the corresponding key–value representations, while the logical decoder computes attention over these representations with its final output used to compute the training loss for gradient updates. The logical encoder is kept frozen during training to ensure cache sharing across tasks. This training procedure, which explicitly accounts for KV cache sharing, helps ensure robustness when KV caches are shared at inference time in real serving scenarios, especially compared with approaches that attempt to share KV caches across models trained independently without considering KV cache sharing.

Figure 2 shows the training loss of LLaMA-3.1-8B on MetaMathQA-40k (Yu et al., 2023) and Evol-Instruct-80k (Roshdieh, 2023). The ICaRus curves almost perfectly overlap with those of conventional task-specific fine-tuning, indicating that restricting learning to the logical decoder does not hinder optimization and is sufficient for task-specific adaptation even when the logical encoder is shared across models. In other words, freezing the logical encoder forces all task-specialized models to reuse a common sequence representation and express their differences only through the decoder, which can be interpreted as a form of implicit regularization.

The core idea of ICaRus is to factorize a decoder-only Transformer into a logical encoder and a logical decoder, and to train only the logical decoder so that KV caches can be shared across different models. Consequently, ICaRus is largely agnostic to how the logical decoder is adapted: in principle, the logical decoder can be trained using full-parameter fine-tuning, LoRA (Hu et al., 2022), or related variants (Liu et al., 2024a; Jiang et al., 2024; Woo et al., 2025). Among these adaptation methods, we adopt LoRA to train the logical decoder because LoRA offers high training efficiency, which enables rapid deployment of new agents in multi-agent systems, while achieving performance comparable to full-parameter fine-tuning (Schulman & Lab, 2025) and making it straightforward to optimize the decoding phase in ICaRus for inference efficiency. In the following section, we describe how we integrate LoRA into ICaRus and how this design further optimizes the overall inference cost.

### 3.3 OPTIMIZING ICARUS FOR MULTI MODEL INFERENCE

In Section 3.2, we introduced the concept and training methodology of ICaRus. In this section, we explain how ICaRus operates in multi model inference scenarios and discuss its key optimization strategies. During the prefill phase, ICaRus uses only the logical encoder, which encodes the input prompt into a KV cache and produces the next token. In the subsequent decode phase, ICaRus duplicates the current token $(x_i)$ and performs two operations: (1) encoding $x_i$ into a key–value pair $(k_i, v_i)$ through the logical encoder, and (2) predicting the task-specific output token $(x_{i+1})$ through the logical decoder by using the duplicated $x_i$ together with the accumulated KV cache $(\{k_1, \ldots, k_i\}, \{v_1, \ldots, v_i\})$, as in Eq. 5. Consequently, regardless of the task, the KV cache is always generated by the logical encoder, and other role-specific decoders can directly reuse this shared KV cache without any need to recompute or further update it. The details can be found in Appendix C

Sequential execution of the logical encoder and decoder may incur up to 2× latency overhead compared to a single model execution, since both weights and KV caches are accessed twice. To mitigate the problem, we insert and fine-tune only lightweight adapters within the logical decoder instead of fully fine-tuning the decoder. Consequently, the logical encoder and logical decoder share most parameters except for the adapters, enabling the shared parameters to be loaded only once and allowing the computations of the two modules to be executed in parallel as depicted in Fig. 3.

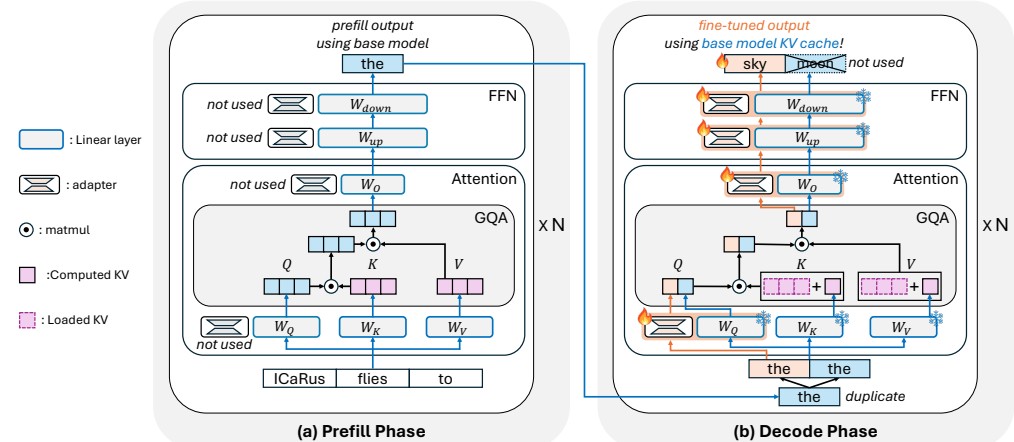

Figure 3: Overview of the ICaRus architecture. The base model, a pretrained decoder-only Transformer, serves as the logical encoder, while the adapter-tuned model (consisting of the base model and a tunable adapter) serves as the logical decoder. The blue and orange lines indicate computations performed by the base model and the adapter-tuned model, respectively. The purple square denotes that the same base model generates the KV cache during both the prefill and decoding phases. In ICaRus architecture, KV caches can be reused regardless of the task, since the KV cache is always generated by the same base model (i.e., the logical encoder).

Table 1: Complexity comparisons between single model and multi model scenarios.

| Scenario | Method | Memory | Latency | | |
| | | | | Decode (per token) | |
| | | Total | Prefill | Memory Access | Compute |
| Single Model | — | $\mathcal{O}(M + L_t)$ | $\mathcal{O}(ML_t + L_t^2)$ | $\mathcal{O}(M + L_t)$ | $\mathcal{O}(M + L_t)$ |
| Multi Model | BaseLine | $\mathcal{O}(M + NL_t)$ | $\mathcal{O}(N(ML_t + L_t^2))$ | $\mathcal{O}(M + L_t)$ | $\mathcal{O}(M + L_t)$ |
| | ICaRus | $\boldsymbol{\mathcal{O}(M + L_t)}$ | $\boldsymbol{\mathcal{O}(ML_t + L_t^2)}$ | $\boldsymbol{\mathcal{O}(M + L_t)}$ | $\mathcal{O}(2M + 2L_t)$ |

In addition, because both models attend to the identical KV cache generated by the base model, we optimize attention computation by concatenating the query representations of the logical encoder and decoder along the head dimension (Fig. 3). This enables parallel attention computation without redundant KV cache reads. Consequently, although the decoding phase of ICaRus appears to double the computational workload by running both the logical encoder and decoder, the system adds only negligible latency overhead. This is because parallel execution generates memory traffic (base parameters, KV caches, and lightweight adapter weights) that is almost the same as that of a single model. The detailed algorithm can be found in Appendix B,

To validate the effectiveness of ICaRus, we further analyze the time and space complexity of multi model system built with the conventional approach (baseline) and with ICaRus, using $N$ adapters in multi agent scenarios. Table 1 summarizes the results. We denote the input prompt length as $L_i$, the number of interaction turns per adapter as $t$, and the number of output tokens per turn as $L_o$, with the total sequence length $L_t = L_i + tL_o$. The base model size is represented by $M$. In the baseline, each model independently allocates KV memory and recomputes prefill for the same prompt, yielding memory $\mathcal{O}(M + NL_t)$ and prefill complexity $\mathcal{O}(N(ML_t + L_t^2))$. In contrast, ICaRus shares a single KV cache across models, reducing both to single model order, with space $\mathcal{O}(M+L_t)$ and prefill $\mathcal{O}(ML_t+L_t^2)$. The advantage grows with longer sequences from inter-model communication and with larger agent counts $N$.

During decoding, the baseline requires $\mathcal{O}(M + L_t)$ memory access and computation per token because each adapter-tuned model reads the model weights and its own KV cache. ICaRus computes both the logical encoder and decoder ($\mathcal{O}(2M + 2L_t)$) but parallelizes most of the computation so

Table 2: Comparison of conventional methods and ICaRus on diverse datasets. Base Model denotes the pretrained decoder-only Transformer without fine-tuning. Multi Model consists of three independently fine tuned models: one on MetaMathQA-40K, one on Evol-Instruct-Code, and one on Oasst1. ICaRus uses the same three specializations, but trains only task-specific logical decoders on a shared logical encoder, enabling KV cache sharing across models.

| Model | Method | KV Sharing | Math | | Coding | | Knowledge |
|-------|--------|------------|------|------|--------|-------|-----------|
| | | | GSM8K | GSM+ | HEval | HEval+ | GPQA |
| LLaMA3.1-8B | Base Model | . | 25.9 | 18.0 | 36.6 | 29.9 | 16.7 |
| | Multi Model | X | **69.7** | **48.5** | 48.2 | 41.5 | 27.3 |
| | ICaRus (Ours) | O | 67.9 | 45.8 | **48.2** | **43.9** | **28.8** |
| Qwen3-8B-Base | Base Model | . | 11.8 | 12.5 | 68.3 | 61.6 | 24.2 |
| | Multi Model | X | 85.4 | 66.1 | 81.7 | 75.6 | **34.3** |
| | ICaRus (Ours) | O | **87.3** | **67.5** | **86.6** | **79.9** | 33.8 |

that the model and KV cache are read only once, restoring $\mathcal{O}(M + L_t)$. In multi-model, long-context, many-turn settings where decoding is memory-bound, memory access dominates; accordingly, ICaRus achieves decoding latency comparable to the baseline.

## 4 EVALUATION

### 4.1 EXPERIMENTAL SETUP

We evaluate ICaRus from two perspectives: (1) accuracy and (2) performance in multi model inference. In section 4.2, we construct multi model systems as follows. Starting from LLaMA-3.1-8B (Dubey et al., 2024) and Qwen3-1.7B/8B/14B-Base (Yang et al., 2025), we build three task-specific models per base model by fine-tuning on MetaMathQA-40k for mathematics (Yu et al., 2023), Evol-Instruct-Code-80k for coding (Roshdieh, 2023), and OASST1 for instruction tuning (Köpf et al., 2023) using either conventional fine-tuning or ICaRus. These systems are then evaluated on benchmarks aligned with each task: GSM8K (Cobbe et al., 2021) and GSM-Plus (Li et al., 2024) for mathematics, HumanEval (Chen et al., 2021) and HumanEval+ (Liu et al., 2023) for coding, and GPQA-Diamond (Rein et al., 2024) for knowledge understanding, using lm-eval-harness (Biderman et al., 2024) and EvalPlus (Liu et al., 2023) to measure zero-shot accuracy. For comparison, both the conventional fine-tuning and ICaRus use LoRA (Hu et al., 2022) as the adaptation method.

For multi model inference (Section 4.3), we measure latency and throughput in a multi agent setting, using representative agentic patterns such as ReAct (Yao et al., 2023) and Reflexion (Shinn et al., 2023) on the HotPotQA dataset (Yang et al., 2018). We evaluate configurations with 2, 4, and 8 agents. We further extend this evaluation to a multi-model, multi-turn request-routing setup: within a single workflow, successive requests from a multi-turn interaction are routed in a round-robin manner to different models. In this setting, the baseline is a conventional multi-LoRA system, whereas ICaRus replaces it with a cache-sharing multi agent system. To ensure a fair comparison, we integrate both systems into the vLLM serving framework and evaluate them under identical settings. More details can be found in the Appendix A.

### 4.2 ACCURACY EVALUATION

**Accuracy on diverse task.**  We first train and evaluate ICaRus alongside conventional fine-tuning across mathematics, coding, and instruction-tuning tasks using LLaMA-3.1-8B and Qwen3-8B, as reported in Table 2. The results show that ICaRus achieves accuracy comparable to, or even surpassing, task-specific fine-tuning across all tasks. In particular, for the Qwen3-8B-Base model, ICaRus outperforms prior task-tuned models by at least 1.4% on benchmark evaluations for both mathematics and coding tasks. We expect that the superior accuracy of ICaRus stems from a generalization effect: by fine-tuning only the logical decoder while keeping the logical encoder frozen, ICaRus reduces the risk of overfitting compared to full task-specific fine-tuning.

Table 3: Comparison of conventional fine-tuning and ICaRus across different model sizes (Qwen3-1.7B/8B/14B-Base) trained on the MetaMathQA-40K dataset.

| Model | Qwen3-1.7B-Base | | Qwen3-8B-Base | | Qwen3-14B-Base | |
|---|---|---|---|---|---|---|
| Method | Baseline | ICaRus | Baseline | ICaRus | Baseline | ICaRus |
| GSM8K | 73.2 | **74.0** | 85.4 | **87.3** | 85.6 | **88.8** |
| GSM+ | 53.7 | **54.1** | 66.1 | **67.5** | 66.7 | **68.8** |

**Scaling with model size.**    We also examine the scalability of ICaRus with respect to model size by conducting experiments on Qwen3-1.7B/8B/14B-Base in Table 3. The results show that ICaRus consistently achieves higher accuracy compared to conventionally fine-tuned baseline, with improvements exceeding 2% on Qwen3-14B-Base, demonstrating that our method remains competitive as model capacity increases. Additionally, we verify the robustness of ICaRus across tasks and its scalability to larger model sizes by evaluating Qwen3-32B on tool-calling tasks, as described in Appendix D.

Table 4: Comparison of conventional methods and ICaRus in multi-model inference scenarios. Base Model denotes the LLaMA-3.1-8B-Base model without fine-tuning, while Math, Coding, and IF denote models fine-tuned on MetaMathQA-40K, Evol-Instruct-Code, and OASST1, respectively. Multi Model and ICaRus both consist of these three task-specific models; in ICaRus, however, only the logical decoders are fine-tuned while the logical encoder is shared across models.

| # Model | Method | KV Sharing | Math | | Coding | | Knowledge | Avg. |
|---|---|---|---|---|---|---|---|---|
| | | | GSM8K | GSM-Plus | HEval | HEval+ | GPQA | |
| 1 | Base Model | . | 25.9 | 18.0 | 36.6 | 29.9 | 16.7 | 25.4 |
| | Math Model | . | **69.7** | **48.5** | 42.7 | 36.6 | 20.7 | 43.6 |
| | Coding Model | . | 22.8 | 17.5 | **48.2** | 41.5 | 21.7 | 30.3 |
| | IF Model | . | 24.5 | 16.5 | 44.5 | 39.0 | 27.2 | 30.3 |
| 3 | Multi Model | X | **69.7** | **48.5** | **48.2** | 41.5 | 27.2 | **47.0** |
| | ICaRus (Ours) | O | 67.9 | 45.8 | **48.2** | 43.9 | 28.8 | 46.9 |

**Multi domain orchestration results.**    Table 4 compares ICaRus orchestration with diverse single and multi model configurations using LLaMA-3.1-8B. Each task-tuned model is fine-tuned on a single domain-specific dataset (MetaMathQA for mathematics, Evol-Instruct-Code-80K for coding, and OASST1 for instruction-tuning). The results show that while a single task-specific fine-tuned model achieves high accuracy on its target task, the model suffers from significant performance degradation on other tasks. In contrast, a multi model system composed of multiple task-specific fine-tuned models achieves consistently high accuracy across all tasks. Our ICaRus also attains accuracy comparable to such multi model system, while additionally benefiting from KV cache sharing across agents, which enables orchestration at substantially lower computational cost.

## 4.3 Performance in Multi Model Inference

**P95 latency and throughput across QPS.**    ICaRus consistently outperforms a baseline multi model system across all load levels in both latency and throughput, as evaluated on LLaMA-3.1-8B under the ReAct pattern (Fig. 4). We measure performance as the number of queries per second (QPS) increases; latency is reported at the 95th percentile (P95).

A key advantage of ICaRus is its ability to reuse identical prefix caches across models, avoiding the redundant recomputation required in baseline system where each model reconstructs its own cache. For example, at QPS 0.3 with 4 models, ICaRus reduces P95 latency by $5.1\times$ compared to the baseline, and this benefit becomes more pronounced as the number of models increases.

As the QPS increases, the cumulative KV cache size of baseline system soon exceeds GPU memory capacity, triggering eviction of previously stored KV caches and their subsequent recomputation. Consequently, throughput first plateaus and then declines, with the degradation occurring earlier as

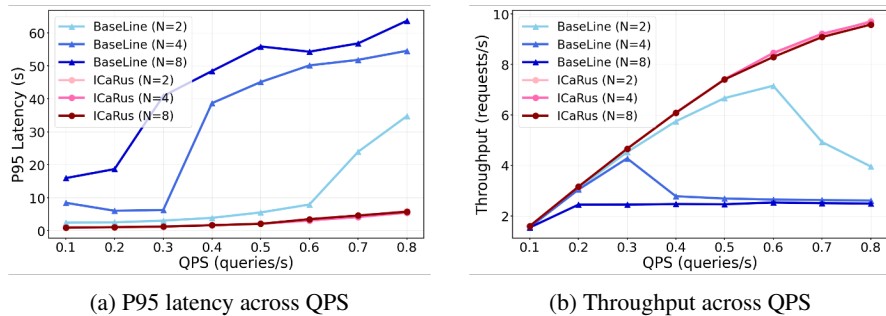

(a) P95 latency across QPS

(b) Throughput across QPS

Figure 4: P95 latency and throughput of ICaRus compared with multiple task-specific agents fine-tuned from the LLaMA-3.1-8B base model under the ReAct pattern. Here, $N$ denotes the number of LoRA modules, which are integrated into multi model system built using either the conventional approach or ICaRus.

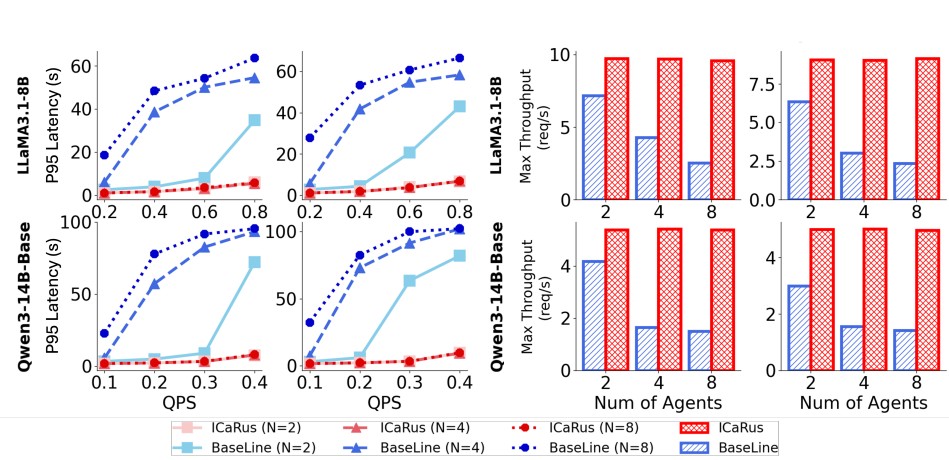

Figure 5: Comparison of P95 latency and maximum throughput across QPS for LLaMA3.1-8B and Qwen-3-14B Base under ReAct and Reflexion patterns.

the number of models increases (e.g., at 0.6 QPS for two models and 0.3 QPS for four models; Fig. 4(b). In contrast, ICaRus avoids redundant cache growth through cross-model KV sharing, allowing throughput to continue increasing even as baseline system plateaus and declines.

Consequently, when comparing maximum achievable throughput, ICaRus outperforms the baseline by $1.4\times$, $2.3\times$, and $3.8\times$ with 2, 4, and 8 models, respectively. At the QPS where baseline system reaches its peak throughput, ICaRus also achieves substantially lower P95 latency-$3.8\times$, $5.1\times$, and $11.1\times$ for 2, 4, and 8 models, respectively. Furthermore, we confirm that ICaRus continues to achieve lower latency and higher throughput than the baseline even in scenarios where evicted KV cache entries are managed by swapping rather than recomputation, as detailed in Appendix E.

**Performance under diverse workflows or models.** We further evaluate baseline system and ICaRus system across different models (LLaMA-3.1-8B and Qwen3-14B-Base) and different agent patterns (ReAct and Reflexion). Specifically, we measure P95 latency over varying QPS and the maximum throughput achieved at the optimal QPS setting, as summarized in Fig. 5.

ICaRus prevents KV cache explosion and enables cross-model prefix caching, thereby achieving lower P95 latency and higher throughput in multi agent workflow. These gains persist even for larger models like Qwen3-14B, where ICaRus achieves up to $7.4\times$ lower latency and $3.6\times$ higher throughput compared to the baseline. Additionally, we verify that the advantages of ICaRus are preserved even under more realistic agentic patterns, where agents are invoked in a random order and the workload is skewed across agents, as demonstrated in Appendix F.

## 5 RELATED WORK

**Multi model Inference** Leveraging multiple models has been widely explored as a way to improve performance over a single model. Routing methods either select the most appropriate model or use multiple models in a cascade (Chen et al., 2024; Shnitzer et al., 2024), while ensemble approaches combine the outputs of multiple models, either at the token level (Yu et al., 2024; Huang et al., 2024) or at the reasoning step level (Park et al., 2025). Multi model approaches have also been applied in multi agent systems, where interactions among agents have been shown to enhance performance across diverse tasks (Fu et al., 2023; Sun et al., 2024; Du et al., 2024). In these systems, each agent used either a base model or fine-tuned variants obtained with methods such as LoRA or instruction tuning (Mineiro, 2024; Liu et al., 2025b).

**KV Cache Optimization** KV cache stores the keys and values of previous tokens to avoid redundant recomputation during autoregressive generation and is traditionally used on a per-request basis (Vaswani et al., 2017). Prefix caching techniques extend the lifetime of the KV cache beyond a single request, enabling multiple turns or related requests to share the same cache (Gao et al., 2024; Gim et al., 2024). However, prefix caching alone cannot address the challenge of deploying multiple models, as KV caches cannot be shared across different models even for identical prompts, and each model generates a distinct KV cache. DroidSpeak (Liu et al., 2025b) addresses this issue by reusing the KV cache of a shared foundational model for non-sensitive layers, while selectively recomputing only the sensitive layers in each agent model. This approach requires identifying sensitive layers that must be recomputed by the agent model, thereby affecting subsequent layers. On a different axis, KVFlow (Pan et al., 2025) manages KV caches by evicting and prefetching based on predetermined agentic workflows instead of an LRU policy, but it remains a single model approach with agents defined by prompts.

## 6 CONCLUSION

In this work, we presented ICaRus, a KV cache-sharing architecture for multi model inference. ICaRus addresses the memory inefficiency of conventional system by enabling cross-model KV cache reuse, while maintaining accuracy through fine-tuning. Experiments across mathematics, coding, and instruction-following tasks confirm that ICaRus delivers accuracy on par with task-specific fine-tuned models, yet achieves significantly lower latency and higher throughput in multi agent workflow. Taken together, these results establish ICaRus as a principled approach for scalable and efficient multi model inference. Looking ahead, we expect ICaRus to extend to large-scale models, heterogeneous agent systems, and real-world deployment scenarios where scalability and efficiency are increasingly critical.

## REPRODUCIBILITY STATEMENT

We formulated the concept of the logical encoder and decoder in detail, which forms the foundation of the ICaRus algorithm, in Section 3.1. Furthermore, we provided a rigorous mathematical formulation of ICaRus, along with its training procedure and convergence of the loss curve, in Section 3.2. The inference process of ICaRus and the corresponding optimization strategies are described in Section 3.3, with pseudocode provided in Appendix B. Finally, the detailed experimental setup for both training and inference is presented in Section 4.1 and Appendix A.

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

## APPENDICES

## A  EXPERIMENTAL SETUP

### A.1  TRAINING SETUP

All experiments were conducted on a single node with $8\times$NVIDIA A100 GPUs (80GB each). Each GPU processed a micro-batch of size 1, and we applied gradient accumulation over 16 steps, resulting in an effective batch size of 128 examples across all devices. This corresponds to approximately 131k tokens per optimization step when the maximum sequence length was 1024, and 262k tokens when it was 2048.

We trained on three datasets: MetaMathQA (40k sampled examples), Evol-Instruct (80k full set), and OASST1 (10k sampled examples). The maximum sequence length was set to 2048 for Evol-Instruct and 1024 for the others. The number of training epochs was 1 for MetaMathQA and Evol-Instruct, and 3 for OASST1.

Optimization was performed using the AdamW optimizer with default hyperparameters ($\beta_1$=0.9, $\beta_2$=0.999) and a weight decay of 0.01. We used a cosine learning rate decay schedule with a warmup ratio of 0.03, and performed a grid search over learning rates $\{1 \times 10^{-4}, 2 \times 10^{-4}, 5 \times 10^{-4}\}$. No additional regularization techniques (e.g., dropout or gradient clipping) were applied.

For all experiments, we applied low-rank adaptation (LoRA) with a rank of 128 and an $\alpha$ of 256.

### A.2  MULTI MODEL INFERENCE SETUP

#### A.2.1  AGENT WORKFLOW SELECTION AND DESIGN

We designed our experimental setup to evaluate the scalability and performance characteristics of multi model AI agent systems under realistic workload conditions. For this study, we selected two representative agentic patterns that reflect common production use cases and exhibit distinct reasoning behaviors:

**ReAct** (Yao et al., 2023): This framework synergizes chain-of-thought reasoning with external tool use through an iterative process where agents generate reasoning traces and task-specific actions in an interleaved manner. In the ReAct paradigm, agents alternate between internal reasoning (thoughts) and external actions (tool calls), with each iteration consisting of a thought-action-observation cycle. This pattern is particularly effective for tasks requiring dynamic interaction with external knowledge bases and APIs.

**Reflexion** (Shinn et al., 2023): This framework reinforces language agents through linguistic feedback, maintaining reflective text in an episodic memory buffer to improve decision-making across multiple trials. Unlike ReAct, Reflexion adds self-evaluation capabilities where agents generate verbal reinforcement cues to assist in self-improvement, storing these experiences in long-term memory for rapid adaptation. This approach enables agents to learn from past mistakes without requiring model fine-tuning, achieving superior performance on complex reasoning tasks.

#### A.2.2  MULTI MODEL ARCHITECTURE WITH LoRA ADAPTERS

To simulate realistic multi-tenant agent deployments, we implemented a multi model inference setup where each agent instance operates with its own Low-Rank Adaptation (LoRA) adapter. This configuration mirrors production scenarios where different agents may require specialized model behaviors or domain-specific fine-tuning. Specifically, we matched the number of concurrent agents to the number of LoRA adapters, ensuring that each agent maintains its own parameter space.

In evaluation, multiple task-specific LoRA adapters share the same base model on a single GPU. Under this setup, both the baseline multi-LoRA system and ICaRus already leverage the standard prefix/KV-aware mechanisms of the serving stack: requests routed to the same LoRA module reuse the existing KV cache for identical prefixes whenever possible, thereby sharing KV-cache memory and avoiding redundant prefill recomputation within each model.

### A.2.3 Workload Characterization

For workload modeling, we used the HotPotQA dataset (Yang et al., 2018) as the underlying question-answering benchmark for both ReAct and Reflexion workflows, following the setup of Kim et al. (2025). Input/output distributions and tool-calling patterns were based on empirical measurements from Kim et al. (2025), which provides comprehensive statistics on real-world agent workflow characteristics. These patterns informed our synthetic workload generation, ensuring our experiments reflect actual deployment scenarios.

### A.2.4 Experimental Parameters

We conducted systematic scaling experiments with the following configuration:

**Agent Scaling**: We evaluated system behavior with 2, 4, and 8 concurrent agents to understand how resource contention and memory pressure evolve with increasing agent density.

**Request Rate (QPS)**:

- For Qwen3-14B-Base: Tested at 0.1, 0.2, 0.3, and 0.4 QPS
- For Llama-3.1-8B: Tested at 0.2, 0.4, 0.6, and 0.8 QPS

The different QPS ranges reflect the computational differences between model sizes, with the smaller 8B model capable of sustaining higher request rates.

**Throughput Measurement**: We measured actual system throughput at the 0.8 QPS configuration to empirically determine system saturation points under peak load conditions.

**Batch Size and Latency Dynamics**: To understand latency behavior under constrained conditions, we fixed the total request count at 128 while varying QPS. This experimental design differs from unbounded request streams where continuously arriving requests would cause monotonically increasing batch sizes and consequently unbounded growth in 95th percentile latency. Under our fixed-request protocol, we observed that 95th percentile latency initially increases with QPS but eventually saturates at a plateau, indicating the system reaches a steady-state where all requests are being processed within the available compute budget.

This saturation behavior provides critical insights into:

- The maximum sustainable batch size for each agent configuration
- The point at which additional request rate increases no longer impact tail latency
- The effective capacity limits of multi agent systems under resource constraints

### A.2.5 Rationale and Implications

Our experimental design captures several critical aspects of production multi agent systems:

1. **Resource Isolation**: By assigning separate LoRA adapters to each agent, we model scenarios where agents require distinct specializations (e.g., different domains, languages, or task-specific fine-tuning).
2. **Memory Pressure**: The multiplicative effect of agent count on KV cache requirements reflects real-world memory bottlenecks in multi-tenant deployments.
3. **Workflow Diversity**: The combination of ReAct's tool-calling patterns and Reflexion's self-improvement cycles represents a broad spectrum of agent behavioral patterns, from reactive tool use to iterative refinement.
4. **Scaling Characteristics**: Our range of agent counts (2–8) and QPS values provides insights into both vertical scaling (request rate) and horizontal scaling (agent parallelism) dimensions.

This setup enables us to quantify the trade-offs between agent autonomy, system throughput, and resource utilization in modern AI agent deployments, providing actionable insights for practitioners deploying multi agent systems at scale.

## B PSEUDO ALGORITHM

### B.1 PREFILL PHASE IN ICARUS

---

**Algorithm 1:** Prefill Phase (Standard Linear Only)

---

**Input:** Prompt tokens $P \in \mathcal{V}^N$
**Output:** First token $y_{\text{prefill}} \in \mathcal{V}$, KV_CACHE$[1 \dots L]$

1   $X_1 \leftarrow \text{Embed}(P) \in \mathbb{R}^{N \times d}$
2   **for** $i = 1$ **to** $L$ **do**
3     $Q_i \leftarrow \text{Linear}(X_i; W_q^i), K_i \leftarrow \text{Linear}(X_i; W_k^i), V_i \leftarrow \text{Linear}(X_i; W_v^i)$
4     $Q_i, K_i \in \mathbb{R}^{N \times d_k}, V_i \in \mathbb{R}^{N \times d_v}$
     /* generate KV cache (w. the Logical Encoder)                */
5     **KV_CACHE**$[i] \leftarrow (\boldsymbol{K_i, V_i})$
6     $A_i \leftarrow \text{Attention}(Q_i, K_i, V_i) \in \mathbb{R}^{N \times d_v}$
7     $X_{i+1} \leftarrow \text{FFN}(\text{AttentionOutput}(A_i)) \in \mathbb{R}^{N \times d}$
8   $y_{\text{prefill}} \leftarrow \text{Sample}(\text{LMHead}((X_{L+1}[N]))$            // Prefill Result

---

## B.2 DECODE PHASE IN ICARUS

---

**Algorithm 2:** ICaRus Linear

---

**Input:** $X \in \mathbb{R}^{2 \times T \times d}$        // batch=2, seqlen $T$, hidden size $d$

1   X[0]: Input for Logical Encoder (Base model)

2   X[1]: Input for Logical Decoder (Base model + Adaptive model)

    **Output:** $Y \in \mathbb{R}^{2 \times T \times d}$

3   /* Parallel execution for Base Model and Adaptive Model        */

4   $\boldsymbol{X}_{\text{temp}} \leftarrow \textbf{Linear}(\boldsymbol{X})$

5   $X_{\text{temp}}[1] \leftarrow X_{\text{temp}}[1] + \text{AdaptiveLinear}(X_{\text{temp}}[1])$

6   $Y \leftarrow X_{\text{temp}}$

---

**Algorithm 3:** Decode Phase (w. ICaRus Linear)

---

**Input:** $y_{\text{prefill}} \in \mathcal{V}$, **KV_CACHE**$[1 \dots L]$

1   KV_CACHE: Prompt KV cache from Logical Encoder (Base Model)

    **Output:** Generated tokens $Y = (y_{N+1}, y_{N+2}, \dots, y_{N+T})$

    (where $N$ is the prompt length, $T$ is the number of generated tokens)

2   $Input\_Token \leftarrow y_{\text{prefill}}$

3   **for** $t = 1 \dots T$ **do**

4      $X_1 \leftarrow \text{Embed}(Input\_Token) \in \mathbb{R}^{N \times d}$

       /* Stack hidden states for ICaRus Execution        */

5      $\boldsymbol{X}_1^{\textbf{pair}} \leftarrow \textbf{stack\_batch}(\boldsymbol{X}_1, \boldsymbol{X}_1)$        // shape: [2,1,d]

6      **for** $i = 1$ **to** $L$ **do**

7          /* KV cache from base model for sharing        */

8          $\boldsymbol{K}_i^{\textbf{step}} \leftarrow \textbf{Linear}(\boldsymbol{X}_i; \boldsymbol{W}_k^i), \boldsymbol{V}_i^{\textbf{step}} \leftarrow \textbf{Linear}(\boldsymbol{X}_i; \boldsymbol{W}_v^i)$

9          $(K_i^{\text{cache}}, V_i^{\text{cache}}) \leftarrow \text{KV\_CACHE}[i]$

10        $K_i \leftarrow \text{concat\_sequence}(K_i^{\text{cache}}, K_i^{\text{step}})$

11        $V_i \leftarrow \text{concat\_sequence}(V_i^{\text{cache}}, V_i^{\text{step}})$

12        KV_CACHE$[i] \leftarrow (K_i, V_i)$

13        $Q_i^{\text{pair}} \leftarrow \text{ICaRusLinear}(X_i^{\text{pair}}; W_q^i, A_q^i)$        // shape: [2,1,H,d_k]

14        /* Enable attention parallelism via GQA        */

15        $\boldsymbol{Q}_i \leftarrow \textbf{concat\_numhead}(\boldsymbol{Q}_i^{\textbf{pair}}[0], \boldsymbol{Q}_i^{\textbf{pair}}[1])$        // shape: [1,2*H,d_k]

16        $\boldsymbol{A}_i \leftarrow \textbf{GQA}(\boldsymbol{Q}_i, \boldsymbol{K}_i, \boldsymbol{V}_i)$        // shape: [1,2*H,d_v]

17        $\boldsymbol{A}_i^{\textbf{pair}} \leftarrow \textbf{transpose\_and\_reshape}(\boldsymbol{A}_i)$        // shape: [2,1,H,d_v]

18        $Z_i^{\text{pair}} \leftarrow \text{ICaRusLinear}(A_i^{\text{pair}}; W_o^i, A_o^i)$        // shape: [2,1,d]

         /* FFN: up $\rightarrow$ act $\rightarrow$ down (W.ICaRusLinear)        */

19        $F_i^{\text{pair}} \leftarrow \text{FFN}(Z_i^{\text{pair}})$        // shape: [2,1,d]

      /* use only Adaptive Result        */

20      $new\_token \leftarrow \text{Sample}(\text{LMHead}(F_{L+1}^{\text{pair}}[1]))$

21      $Y \leftarrow \text{concat}(Y, new\_token)$

22      $Input\_Token \leftarrow new\_token$

---

## C  Logical Encoder–Decoder: Concept and Inference Workflow

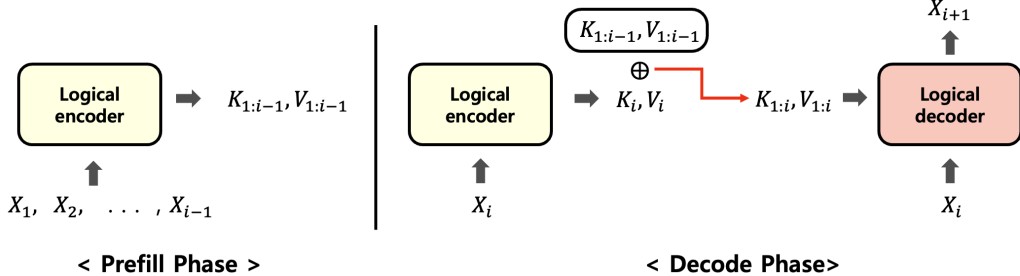

Figure 6: Inference workflow of the logical encoder-decoder.

In this section, we provide a more detailed explanation of the logical encoder–decoder concept. Inference in a decoder-only Transformer can be viewed as consisting of two phases: a *prefill* phase and a *decode* phase.

- **Prefill:** generate the KV cache for the input prompt.
- **Decode:** (1) generate the KV cache for the current token, and (2) predict the next token.

Motivated by this behavior, we conceptually decompose the model into a *logical encoder* and a *logical decoder*. The logical encoder denotes the part of the computation that is solely responsible for producing the KV cache, whereas the logical decoder denotes the part that predicts the next token during decoding and does not produce any new KV entries: it treats the KV cache as a pre-computed sequence representation and only issues queries against it to generate tokens. Under this decomposition, inference can be reinterpreted as follows:

- **Prefill:** the logical encoder generates the KV cache for the input prompt.
- **Decode:** (1) the logical encoder generates the KV cache for the current token, and (2) the logical decoder predicts the next token.

ICaRus fine-tunes only the logical decoder and freeze logical encoder. Specifically, the task-specialized decoders consume the shared KV cache from the common logical encoder for attention computation, as shown in Fig. 6, enabling heterogeneous, task-specialized decoders to operate on a single shared representation without any approximation or recomputation. In other words, ICaRus models can reuse KV cache entries produced not only in the prefill phase but also in the decode phase without any updates or reconstruction, because all KV entries are always generated by the same logical encoder.

## D  Robustness of ICaRus on Tool-Calling Tasks with Larger Models

To demonstrate the scalability and robustness of ICaRus, We conducted experiments with Qwen3-32B on the ToolAce dataset (Liu et al., 2025a) for tool calling related task, and evaluated the resulting models on the BFCL benchmark as shown below.

As shown in Fig. 7, the loss curve of ICaRus converges smoothly and is comparable to that of the baseline, which is consistent with the behavior observed in Fig. 2 of the manuscript for math and coding tasks with 8B models. This indicates that our training procedure remains stable even when scaling to larger models and to a different task domain.

Moreover, as reported in Table 5, even with a larger 32B model and the tool calling task, ICaRus achieves comparable accuracy than a baseline that does not share the KV cache. This suggests that our method is not only trainable and stable, but also robust and effective, both in terms of model scale and task type.

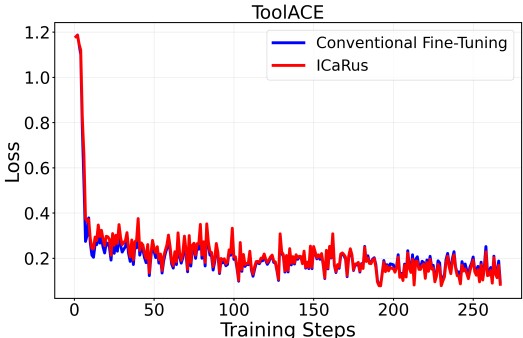

Figure 7: Training loss curves of conventional fine-tuning and ICaRus, both applied with LoRA on Qwen-3-32B, trained on the ToolAce dataset.

Table 5: Comparison of conventional fine-tuning and ICaRus when training Qwen3-32B on the ToolAce dataset.

| Model | Method | BFCL Non-live (AST) | | |
|---|---|---|---|---|
| | | Simple Python | Simple Java | Simple JavaScript |
| Qwen3-32B | Baseline | **96.5** | 62.0 | 74.0 |
| | ICaRus (Ours) | 94.5 | **63.0** | **76.0** |

# E  ICARUS UNDER SWAP-BASED KV CACHE MANAGEMENT

We conducted experiments with swap enabled (4GB swap space) using an earlier version of vLLM that supports this feature. The experimental results are reported below.

Figure 8 shows that ICaRus continues to provide lower P95 latency and higher throughput even when the multi-model system uses swap for KV cache management. In particular, with 8 LoRA modules, ICaRus achieves up to 12.1× lower P95 latency and 3.8× higher throughput than the baseline. This is because ICaRus reduces the KV cache footprint itself, so that even at higher QPS the GPU does not saturate and expensive swap operations are rarely triggered in the first place.

In summary, we emphasize that recompute/swap strategies and ICaRus address orthogonal aspects of the problem. Concretely, recompute or swap determine how to manage KV cache once GPU memory becomes full (e.g., whether to evict and reload from host storage or to recompute), whereas ICaRus fundamentally reduces KV pressure by enabling cross-model KV sharing across task-specialized models. By avoiding redundant KV construction across models, ICaRus effectively

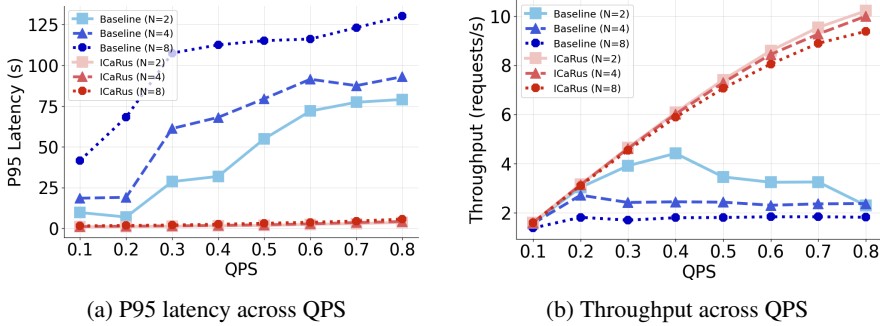

(a) P95 latency across QPS          (b) Throughput across QPS

Figure 8: P95 latency and throughput of ICaRus compared with multiple task-specific agents fine-tuned from the LLaMA-3.1-8B base model under the ReAct workflow with swap-based KV cache management. Here, $N$ denotes the number of LoRA modules, which are integrated into multi model system built using either the conventional approach or ICaRus.

delays or mitigates the point at which the KV cache saturates GPU memory, thereby improving performance regardless of whether the underlying system chooses recompute or swap as its eviction policy. In principle, ICaRus could also be combined with swap-based KV management.

## F  PERFORMANCE UNDER RANDOM AND SKEWED AGENTIC PATTERN IN REAL-WORLD SCENARIOS

We evaluate the scenario in which the controller invokes agents at random with a skewed workload under ReAct workflow, so that on a typical turn only a subset of agents is active, better reflecting such real-world scenarios. Specifically, unlike the round-robin invocation pattern in Section 4.3, we construct a skewed workload in which one agent is invoked with probability 50% on each turn, while the remaining agents share the rest of the probability mass and are invoked in a random order rather than a fixed sequence. The experiments are conducted on the vLLM v0 architecture and the results are reported below.

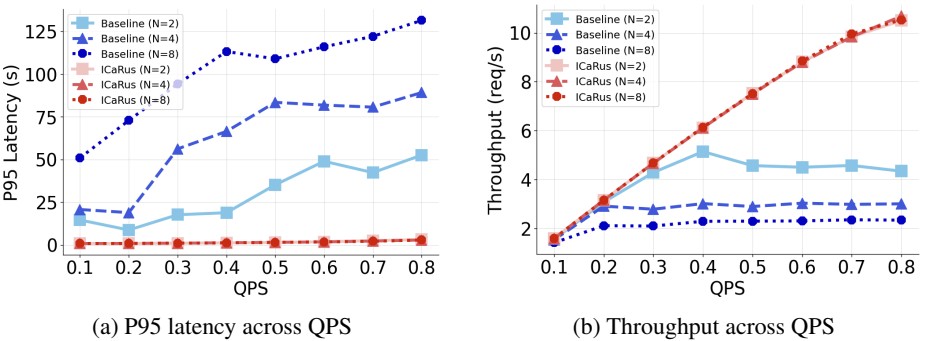

(a) P95 latency across QPS          (b) Throughput across QPS

Figure 9: P95 latency and throughput of ICaRus compared with multiple task-specific agents fine-tuned from the LLaMA-3.1-8B base model under the ReAct workflow where the agent invocation pattern is random and skewed. Here, $N$ denotes the number of LoRA modules, which are integrated into multi model system built using either the conventional approach or ICaRus.

Fig. 9 shows that ICaRus maintains low P95 latency and high throughput under dynamic and skewed agentic patterns. For example, with 2 models at 0.4 QPS, ICaRus achieves 15× lower P95 latency and 1.2× higher throughput than the baseline, demonstrating that the core advantage of ICaRus, enabling per-model prefix caching on top of cross-model KV sharing, is preserved even under skewed and random agent invocation patterns. Furthermore, in the baseline, throughput quickly saturates beyond a certain QPS because rapid growth of the KV cache triggers frequent evictions and recomputations. In contrast, ICaRus allows multiple models to share a single KV cache pool, keeping entries within the available GPU memory budget without eviction so that throughput continues to increase with QPS without saturation. As a result, in the 8-model setting, ICaRus achieves up to 3.5× higher throughput than the baseline under skewed and dynamic agent invocation patterns.

