# OpenReview forum: "ICaRus: Identical Cache Reuse for Efficient Multi-Model Inference"
_ICLR.cc/2026/Conference — ICLR 2026 Poster_

### Official Review · Reviewer_QNML · 2025-10-16

**Soundness:** 3
**Presentation:** 3
**Contribution:** 2
**Rating:** 6
**Confidence:** 4

**Summary:**

This paper introduces ICaRus, a novel architecture designed to optimize multi-model inference by enabling full Key-Value (KV) cache sharing across decoder-only Transformer models. The core innovation lies in conceptually decomposing a Transformer into a logical encoder (which generates KV caches) and a logical decoder (which predicts output tokens). By freezing the encoder and fine-tuning only the decoder using lightweight adapters (e.g., LoRA), ICaRus allows multiple specialized models to reuse identical KV caches, significantly reducing memory usage and recomputation overhead. The authors demonstrate that ICaRus achieves comparable accuracy to task-specific fine-tuned models while delivering up to 11.1× lower P95 latency and 3.8× higher throughput in multi-agent workflows.

**Strengths:**

* The logical encoder-decoder decomposition is an interesting and insightful abstraction that enables cross-model KV cache reuse, a previously unsolved challenge.
* ICaRus shows substantial improvements in latency and throughput, especially in multi-agent workflows like ReAct and Reflexion, while surprisingly matches or even exceeds the performance of fully fine-tuned models in several benchmarks.

**Weaknesses:**

* Limited scope of sharing. The proposed approach assumes problems are solved with multiple task-specialized models that **have different weights** and **are finetuned from the same base model**, such an assumption limits its applications. Is such a setting common? To my best knowledge, multi-agent systems typically use the same model for all roles, with only the system prompts customized for each.

* Synthetic evaluation setup. The experimental setting for multi-model inference (Lines 318–320) relies on a synthesized workload that routes multi-turn requests in a round-robin fashion to different models. While this setup enables controlled benchmarking, it may not accurately reflect the dynamic and heterogeneous nature of real-world multi-agent systems, where agent interactions are often asynchronous, task-dependent, and influenced by external stimuli. This raises concerns about the generalizability of the reported latency and throughput improvements.

**Questions:**

1. In Figure 3, does the decode phase update the **shared** KV Cache? If yes, does another role-specific decoder directly reuse the updated KV or it always requires the logical encoder (Figure 3(a)) to recompute the KV cache?

2. Is ICaRus coupled with LoRA finetuning? Or it is compatible with other finetuning approaches (e.g., full parameter finetuning or prefix tuning)? Which adaptations are needed for other finetuning approaches?

---

> ### Author Response · Authors · 2025-11-19
> **Response to Reviewer QNML**
>
> **Q1.** Limited scope of sharing. The proposed approach assumes problems are solved with multiple task-specialized models that have different weights and are finetuned from the same base model, such an assumption limits its applications. Is such a setting common? To my best knowledge, multi-agent systems typically use the same model for all roles, with only the system prompts customized for each.
>
>
> **A1.** We appreciate this observation and agree that many early multi-agent systems follow the pattern you describe, reusing a single base model across roles and distinguishing them only through different system prompts. Prompt engineering is a powerful and widely adopted technique, and our approach is compatible with such deployments. However, we would like to emphasize that **there is a clear shift in both research and practice toward using multiple task or domain-specialized models.**
>
> Several works have shown that **multi-agent systems can substantially outperform single-agent systems [1, 2]**, and **performance improves as the number of fine-tuned agents increases [2]**. Industrial efforts follow a similar direction: for example, **NVIDIA [3]** has highlighted multi-agent systems composed of multiple fine-tuned small language models as a key part of the future of agentic AI, and products such as **Microsoft’s Copilot Studio [4]** are beginning to adopt this paradigm in real-world applications. Yet in these multi-agent scenarios, the inability to reuse KV caches across models leads to significant memory and computation overheads during inference. We believe that **ICaRus offers a principled way to remove this KV-cache bottleneck in emerging multi-agent, multi-model serving workloads, thereby making agentic AI systems more scalable and practical in real deployments.**

---

> ### Author Response · Authors · 2025-11-19
> **Response to Reviewer QNML**
>
> **Q2.** Synthetic evaluation setup. The experimental setting for multi-model inference (Lines 318–320) relies on a synthesized workload that routes multi-turn requests in a round-robin fashion to different models. While this setup enables controlled benchmarking, it may not accurately reflect the dynamic and heterogeneous nature of real-world multi-agent systems, where agent interactions are often asynchronous, task-dependent, and influenced by external stimuli. This raises concerns about the generalizability of the reported latency and throughput improvements. (Appendix)
>
> **A2.**
>
> ---
>
> + **Round-Robin Pattern Examples in Real Systems**
>
> Thank you for pointing this out. Our multi-model evaluation is to capture a representative case where several agents are invoked on the same conversation context within a turn. **This style of round-robin, multi-agent interaction appears in many existing systems.** For example, the multi-agent debate pattern in **AutoGen [5]** runs multiple solver agents plus an aggregator on the same task, allowing them to take turns responding and to refine or compete with each other. Similarly, **MetaGPT [6]** constructs a software-engineering workflow in which role-specific agents (planning, architecture design, system design, coding, etc.) are invoked sequentially for a single user request, all sharing an evolving context. To reflect the prevalence of such round-robin interactions in practical multi-agent systems, **prior multi-agent inference works such as DroidSpeak [7] and KVFlow [8] also evaluate their methods under this representative, widely used round-robin pattern.** Following this line of work, our round-robin multi-LoRA setting models workflows in which a fixed set of agents is invoked sequentially on a shared prefix.
>
> ---
>
> + **Additional Experimental: Random and Skewed Agent Invocation Patterns**
>
> That said, we fully agree that real-world multi-agent systems are more dynamic and heterogeneous than our idealized round-robin setup. **To partially bridge this gap and test the generalizability of our latency/throughput gains, we ran an additional experiment** where a ReAct-style controller invokes agents **at random under a skewed workload.** Specifically, unlike the round-robin invocation pattern in the manuscripts, we construct a skewed workload in which one agent is invoked with probability 50% on each turn, while the remaining agents share the rest of the probability mass and are invoked in a random order rather than a fixed sequence. The experiments are conducted on the vLLM v0 architecture and the results are reported below.
>
> > [Figure R1.1 P95 latency and throughput of ICaRus under the ReAct workflow where the agent invocation pattern is random and skewed.](https://drive.google.com/file/d/1ZmQupmfbd1Y2FlCYcaC_AYRFY7sDhGiv/view?usp=sharing)
>
> Figure R1.1 shows that ICaRus maintains low P95 latency and high throughput under dynamic and skewed agentic patterns. For example, with 2 models at 0.4 QPS, ICaRus achieves **15× lower P95 latency** and **1.2× higher throughput** than the baseline, demonstrating that **the core advantage of ICaRus, enabling per-model prefix caching on top of cross-model KV sharing,** is preserved even under skewed and random agent invocation patterns. Furthermore, in the baseline, throughput quickly saturates beyond a certain QPS because rapid growth of the KV cache triggers frequent evictions and recomputations. In contrast, ICaRus allows multiple models to share a single KV cache pool, **keeping entries within the available GPU memory budget without eviction** so that throughput continues to increase with QPS without saturation. As a result, in the 8-model setting, ICaRus achieves up to **3.5× higher throughput** than the baseline under skewed and dynamic agent invocation patterns.
>
> ---
>  We thank the reviewer for this insightful suggestion, which led us to confirm that ICaRus enjoys even larger advantages over conventional systems under realistic dynamic and skewed agentic workloads. We incorporated these results into the Appendix F of the revised manuscripts.

---

> ### Author Response · Authors · 2025-11-19
> **Response to Reviewer QNML**
>
> **Q3.** In Figure 3, does the decode phase update the shared KV Cache? If yes, does another role-specific decoder directly reuse the updated KV or it always requires the logical encoder (Figure 3(a)) to recompute the KV cache?
>
> **A3.** We apologize for the confusion and thank the reviewer for pointing this out. **We clarify that the KV cache produced during the decode phase is also shared across all models as decode phase updates the shared KV Cache.**
>
> Concretely, in the **decode phase** (Figure 3(b) of the manuscripts), two operations happen simultaneously in ICaRus:
>
> > **The logical encoder computes the KV cache for the current token** and appends it to the shared KV cache
>  (blue line in Figure 3(b)).
>
>
> > **The logical decoder then predicts the next token** by reusing the shared KV cache produced by the logical encoder in both the prefill and decode phases
>  (orange line in Figure 3(b)).
>
> Therefore,  the KV cache is always generated by the logical encoder, and other role-specific decoders can directly reuse this shared KV cache even  **when it is generated during the decoding phase, without any need to recompute or further update it.** We also parallelize these two operations for improving efficiency, as illustrated in Figure 3 and summarized in Table 1 (Section 3.3 of the manuscript).
>
> We appreciate the reviewer for raising this source of ambiguity and revised the paper to state this behavior more clearly in the caption of Figure 3, Section 3.3, and Appendix C.

---

> ### Author Response · Authors · 2025-11-19
> **Response to Reviewer QNML**
>
> **Q4.** Is ICaRus coupled with LoRA finetuning? Or it is compatible with other finetuning approaches (e.g., full parameter finetuning or prefix tuning)? Which adaptations are needed for other finetuning approaches?
>
> **A4.**
>
> ---
> + **ICaRus Is Orthogonal to the Choice of Fine Tuning Method**
>
> We thank the reviewer for this interesting question. We emphasize that **ICaRus is not coupled to LoRA, and it is compatible with other finetuning approaches** (including full parameter finetuning and prefix/prompt tuning), by **applying these training techniques to train the logical decoder** instead of LoRA. ICaRus builds on the observation that a decoder-only Transformer can be conceptually decomposed into a logical encoder, which constructs the KV cache, and a logical decoder, which consumes this cache to predict the next token. In ICaRus, we **freeze the logical encoder and finetune only the logical decoder**, so that multiple task-specialized decoders (math, coding, instruction, etc.) share the same encoder and therefore the same KV cache for a given prefix. In other words, the crucial aspect of ICaRus is that it fine tunes only the logical decoder; **the specific adaptation method used to train it, such as full parameter fine tuning, prompt tuning, or LoRA, is orthogonal to our approach.**
>
> ---
>
> + **Why We Use LoRA to Instantiate the Logical Decoder**
>
> In our experiments, we instantiate the task-specific logical decoders with LoRA adapters because LoRA is a widely adopted, efficient, and high-quality PEFT method comparable with full parameter finetuning [9]. However, **ICaRus does not rely on LoRA in any essential way.** Any adaptation scheme that modifies only the logical decoder parameters—such as DoRA, prefix-/prompt-tuning, or even **full-parameter finetuning of the logical decoder**—is compatible with our framework, provided that the logical encoder remains shared and frozen.
>
> In summary, the core novelty of ICaRus lies in **the encoder/decoder factorization and the identical-cache property across models,** rather than in the particular choice of LoRA. We have added a clarifying remark in Section 3.2 of the revised manuscript to explicitly state this compatibility with other finetuning approaches.

---

> ### Author Response · Authors · 2025-11-19
> **Response to Reviewer QNML**
>
> [1] Shen et al., “Small LLMs Are Weak Tool Learners: A Multi-LLM Agent”, EMNLP 2024.
>
> [2] Subramaniam et al., “Multiagent Finetuning: Self Improvement with Diverse Reasoning Chains”, ICLR 2025.
>
> [3] Belcak et al., “Small Language Models Are the Future of Agentic AI”, arXiv 2025.
>
> [4] Spataro, “Introducing Microsoft 365 Copilot Tuning, Multi-Agent Orchestration, and More from Microsoft Build 2025”, Microsoft 365 Blog, 2025.
>
> [5] Wu et al., “AutoGen: Enabling Next-Gen LLM Applications via Multi-Agent Conversation Framework”, COLM 2024.
>
> [6] Hong et al., “MetaGPT: Meta Programming for A Multi-Agent Collaborative Framework”, ICLR 2024.
>
> [7] Liu et al., “DroidSpeak: KV Cache Sharing for Cross-LLM Communication and Multi-LLM Serving”, arXiv 2024.
>
> [8] Pan et al., “KVFlow: Efficient Prefix Caching for Accelerating LLM-Based Multi-Agent Workflows”, NeurIPS 2025.
>
> [9] Schulman & Thinking Machines Lab, “LoRA Without Regret”, Thinking Machines Lab: Connectionism, 2025.

---

> ### Author Response · Authors · 2025-11-27
> **Gentle Reminder regarding our Rebuttal**
>
> Dear Reviewer QNML,
>
> We would like to once again sincerely thank you for your careful review and positive evaluation of our work. As the discussion period is drawing to a close, we would like to respectfully remind you that we have thoroughly revised the manuscript to reflect all of your insightful comments, and we summarize the main changes in the revised version below.
>
> + In our response A1, we use several concrete examples to show that, in both industry and academia, multi-agent systems are undergoing a **paradigm shift from using the same model for all roles to employing multiple task-specialized models** in order to build higher-quality agentic systems.
>
> + In our response A2, we **conducted experiments under random and skewed agent Invocation patterns** to better reflect real-world scenarios and confirmed that ICaRus **achieves higher performance** than existing systems even in these settings; these additional results and analyses have been incorporated **into Appendix F of the revised version.**
>
> + In our response A3, we clarify that **the KV cache produced during the decoding phase is also shared across all models**, since the decoding phase updates the shared KV cache; this clarification has been reflected **in the caption of Figure 3, Section 3.3, and Appendix C of the revised manuscript.**
>
> + In our response A4, we emphasize that ICaRus **is orthogonal to the choice of fine-tuning method**, and this point has been reflected **in Section 3.2 of the revised manuscript.**
>
> We are very grateful for your constructive feedback, which has significantly improved the quality of the paper, and we hope that the revised version and our responses have adequately addressed your concerns. We look forward to your further feedback.
>
> Best regards,
>
> The Authors

---

> ### Comment · Reviewer_QNML · 2025-11-27
> **Response to the authors' rebuttal**
>
> Thank you for the detailed clarifications and additional experiments provided in your rebuttal. The rebuttal addresses my main concerns regarding scope, evaluation realism, and technical details. The additional experiments and clarifications improve the robustness of the claims. I will raise my score to 8.

---

### Official Review · Reviewer_eTzX · 2025-10-27

**Soundness:** 2
**Presentation:** 3
**Contribution:** 3
**Rating:** 6
**Confidence:** 3

**Summary:**

ICaRus introduces “Identical Cache Reuse” to let multiple task-specialized LLMs share the same KV cache for identical prompts, avoiding redundant memory use and recomputation in multi-model inference. It splits a decoder-only Transformer into a frozen logical encoder (that builds KV caches) and a fine-tuned logical decoder (that predicts tokens), so all models reuse one shared cache. Using lightweight adapters (e.g., LoRA), ICaRus achieves similar accuracy to task-specific fine-tuning but cuts P95 latency by up to 11× and boosts throughput up to 3.8× in multi-agent workflows.

**Strengths:**

1. Innovative idea that changes how multi-model share KV cache.
2. Demonstrates good potential system improvement over previous methods by the new design.
3. Promising path to efficient multi-model inference.

**Weaknesses:**

1. Unclear accuracy comparison to workflows without any sharing or naive baselines. How does a base model perform if I do not use the new workflow?
2. Training seems to be more complex. How robust is this training method? It will be more convincing to do more experiments on bigger models and more datasets.

**Questions:**

1. How does the work deal with cases when agent finishes prefill and does some decoding? How to reuse the decode token for that part? Or how does system perform if that part is not used? Especially in long decoding scenarios like coding.

2. How does the accuracy compare with not sharing at all? A simpler workflow without multi-model inference. Not sure if the dataset needs multi-model inference.

---

> ### Author Response · Authors · 2025-11-19
> **Response to Reviewer eTzX**
>
> **Q1.** Unclear accuracy comparison to workflows without any sharing or naive baselines. How does a base model perform if I do not use the new workflow? How does the accuracy compare with not sharing at all? A simpler workflow without multi-model inference. Not sure if the dataset needs multi-model inference.
>
> **A1.** We apologize for the confusion caused by unclear wording in the manuscript. As requested by the reviewer, **we have already reported in Section 4.2 (Tables 2 and 4) the naive baseline that does not require KV-cache sharing (the single-model setting).** The corresponding results are as follows.
>
> > Table R3.1. Comparison of conventional methods and ICaRus on diverse datasets. Single Model denotes the base model without fine tuning. Multi Model consists of three independently fine tuned models: one on MetaMathQA-40K, one on Evol-Instruct-Code, and one on Oasst1. ICaRus uses the same three specializations, but trains only task-specific logical decoders on a shared logical encoder, enabling KV cache sharing across models.
>
> | Model          | Method         | KV Sharing | GSM8K (Math) | GSM+ (Math) | HEval (Coding) | HEval+ (Coding) | GPQA (Knowledge) |
> |----------------|----------------|------------|--------------|-------------|----------------|-----------------|------------------|
> | LLaMA3.1-8B    | Single Model   | .          | 25.9         | 18.0        | 36.6           | 29.9            | 16.7             |
> |     | Multi Model    | X          | **69.7**     | **48.5**    | 48.2           | 41.5            | 27.3             |
> |     | ICaRus (Ours)  | O          | 67.9         | 45.8        | **48.2**       | **43.9**        | **28.8**         |
> | Qwen3-8B-Base  | Single Model   | .          | 11.8         | 12.5        | 68.3           | 61.6            | 24.2             |
> |   | Multi Model    | X          | 85.4         | 66.1        | 81.7           | 75.6            | **34.3**         |
> |   | ICaRus (Ours)  | O          | **87.3**     | **67.5**    | **86.6**       | **79.9**        | 33.8             |
>
> > Table R3.2. Comparison of conventional methods and ICaRus in multi-model inference scenarios. Base Model denotes the LLaMA-3.1-8B-Base model without fine-tuning, while Math, Coding, and IF denote models fine-tuned on MetaMathQA-40K, Evol-Instruct-Code, and OASST1, respectively. Multi Model and ICaRus both consist of these three task-specific models; in ICaRus, however, only the logical decoders are fine-tuned while the logical encoder is shared across models.
>
> | # Model | Method        | KV Sharing | GSM8K (Math) | GSM-Plus (Math) | HEval (Coding) | HEval+ (Coding) | GPQA (Knowledge) | Avg. |
> |--------|---------------|------------|--------------|-----------------|----------------|-----------------|------------------|------|
> | 1      | Base Model    | .          | 25.9         | 18.0            | 36.6           | 29.9            | 16.7             | 25.4 |
> |       | Math Model    | .          | **69.7**     | **48.5**        | 42.7           | 36.6            | 20.7             | 43.6 |
> |       | Coding Model  | .          | 22.8         | 17.5            | **48.2**       | 41.5            | 21.7             | 30.3 |
> |       | IF Model      | .          | 24.5         | 16.5            | 44.5           | 39.0            | 27.2             | 30.3 |
> | 3      | Multi Model   | X          | **69.7**     | **48.5**        | **48.2**       | 41.5            | 27.2             | **47.0** |
> |       | ICaRus (Ours) | O          | 67.9         | 45.8            | **48.2**       | **43.9**        | **28.8**         | 46.9 |
>
> We confirm in Table R3.1 and Table R3.2 that **ICaRus achieves higher accuracy than the naive single-model baseline across various datasets and base models,** as multiple task-specific experts fine-tuned for different tasks can collaborate within a single system. This observation is consistent with **prior work [1, 2] showing that systems composed of several fine-tuned models or agents often outperform simple prompting of a single model.** We again apologize for the earlier lack of clarity and have updated Tables 2 and 4 in the main text to explicitly indicate whether each setting uses a single- or multi-model configuration and whether KV cache sharing is enabled.

---

> ### Author Response · Authors · 2025-11-19
> **Response to Reviewer eTzX**
>
> **Q2.** Training seems to be more complex. How robust is this training method? It will be more convincing to do more experiments on bigger models and more datasets.
>
> **A2.** We fully agree with the reviewer that, given the seemingly more complex training, it is important to demonstrate the robustness of our method on **larger models and additional datasets**.
>
> To address this concern, we **conducted new experiments with Qwen3-32B on the ToolAce dataset** for tool calling tasks, and evaluated the resulting models on the BFCL benchmark.
>
> > [Figure R3.1. Training loss curves of conventional fine-tuning and ICaRus, both applied with LoRA on Qwen-3-32B, trained on the ToolAce dataset.](https://drive.google.com/file/d/1ZmQupmfbd1Y2FlCYcaC_AYRFY7sDhGiv/view?usp=sharing)
>
> > Table R3.3. The comparisons of conventional fine-tuning (baseline) and ICaRus in Qwen3-32B trained on ToolAce datasets.
>
> |                          |  |   BFCL-nonlive    |           |
> |--------------------------|--------------|-------|-----------|
> |                          | Simple Python       | Simple Java  | Simple JavaScript|
> | Baseline | **96.5**   | 62.0  | 74.0      |
> | ICaRus                   | 94.5         | **63.0** | **76.0** |
>
> As shown in Figure R3.1, **the loss curve of ICaRus converges smoothly and is comparable to that of the baseline,** which is consistent with the behavior observed in Figure 2 of the manuscript for math and coding tasks with 8B models. This indicates that our training procedure **remains stable even when scaling to larger models and to a different task domain.**
>
> Moreover, as reported in Table R3.1, even with a larger 32B model and the tool calling task, ICaRus **achieves comparable accuracy than a baseline** that does not share the KV cache. This suggests that our method is not only trainable and stable, but also **robust and effective, both in terms of model scale and task type.**
>
> We thank the reviewer again for raising this important point, and we have incorporated these additional results and the corresponding discussion into the Appendix D of the revised manuscript.

---

> ### Author Response · Authors · 2025-11-19
> **Response to Reviewer eTzX**
>
> **Q3.** How does the work deal with cases when agent finishes prefill and does some decoding? How to reuse the decode token for that part? Or how does system perform if that part is not used? Especially in long decoding scenarios like coding.
>
> **A3.** We apologize for the confusion and thank the reviewer for pointing this out. **We clarify that the KV cache produced during the decode phase is also shared across all models.**
>
> Concretely, in the **decode phase** (Figure 3(b) of the manuscripts), two operations happen simultaneously in ICaRus:
>
> > **The logical encoder computes the KV cache for the current token** and appends it to the shared KV cache
>  (blue line in Figure 3(b)).
>
>
> > **The logical decoder then predicts the next token** by reusing the shared KV cache produced by the logical encoder in both the prefill and decode phases
>  (orange line in Figure 3(b)).
>
> Therefore,  the KV cache is always generated by the logical encoder, and other role-specific decoders can directly reuse this shared KV cache even  **when it is generated during the decoding phase, without any need to recompute or further update it.** We also parallelize these two operations for improving efficiency, as illustrated in Figure 3 and summarized in Table 1 (Section 3.3 of the manuscript).
>
> We appreciate the reviewer for raising this source of ambiguity and revised the paper to state this behavior more clearly in the caption of Figure 3, Section 3.3, and Appendix C.

---

> ### Author Response · Authors · 2025-11-19
> **Response to Reviewer eTzX**
>
> [1] Shen et al., “Small LLMs Are Weak Tool Learners: A Multi-LLM Agent”, EMNLP 2024.
>
> [2] Subramaniam et al., “Multiagent Finetuning: Self Improvement with Diverse Reasoning Chains”, ICLR 2025.

---

> > ### Comment · Reviewer_eTzX · 2025-11-19
> >
> > Makes sense to me. Please add experiments to final paper and explain how the decoding time after prefill affects the workload.

---

> > > ### Author Response · Authors · 2025-11-27
> > > **Response to Reviewer eTzX**
> > >
> > > We sincerely thank the reviewer for their insightful comments and for recognizing the contribution of our work, which led to raising the score from 6 to 8.
> > >
> > > As shown in Table 1, our analysis indicates that the amount of memory access during the decoding phase in ICaRus is comparable to that of the baseline, while the overall system throughput, including both prefill and decoding, is significantly improved, as discussed in Section 4. This improvement comes from fully sharing across multiple models not only the input prompts but also the KV cache produced during the decoding phase.
> > >
> > > Following the reviewer’s suggestion, we will revise the final version to report prefill time and decoding time separately for each workload, so that the impact on each workload is clearly reflected.

---

### Official Review · Reviewer_1HaB · 2025-10-31

**Soundness:** 3
**Presentation:** 3
**Contribution:** 2
**Rating:** 4
**Confidence:** 4

**Summary:**

* The paper proposes a novel method to reuse KV Caches across different models to reduce redundant prefill operations. The method achieves up to 11.1× lower P95 latency and 3.8× higher throughput in agentic workflow across 8 different models.

**Strengths:**

* The method decouples the decoder-only model into a logical encoder and a token predictor, and freezes the encoder for fine-tuning. This makes the fine-tuning faster and more memory efficient. All LLMs also share the same logical encoder.
* Instead of fine-tuning the entire decoder, lightweight adapters are used for fine-tuning for more efficiency to reduce the overhead of multiple KV accesses.

**Weaknesses:**

* The paper mentions DroidSpeak as related work, which also performs sharing of the KV Cache across multiple LLMs, but points out that DroidSpeak requires some re-computation of sensitive layers. It would be useful to see a more direct comparison with DroidSpeak as a baseline and compare the cost of fine-tuning adapters in ICaRus vs selective re-computation in DroidSpeak.
* The evaluation only considers a round-robin routing scheme, which deliberately increases the number of models being used in a session. It would be useful to see the evaluation under prefix-aware or KV-aware routing. Do the gains still hold in terms of the latency and throughput if reuse across models is not required?

**Questions:**

* In the round-robin evaluation setup, how many different models are used? I missed this detail and would encourage the authors to clarify.
* In Figure 2, please clarify the insight as the training loss curves look almost identical and I am missing the takeaway.

---

> ### Author Response · Authors · 2025-11-19
> **Response to Reviewer 1HaB**
>
> **Q1.** The paper mentions DroidSpeak as related work, which also performs sharing of the KV Cache across multiple LLMs, but points out that DroidSpeak requires some re-computation of sensitive layers. It would be useful to see a more direct comparison with DroidSpeak as a baseline and compare the cost of fine-tuning adapters in ICaRus vs selective re-computation in DroidSpeak.
>
> **A1.** Thank you for this insightful question. Following the reviewer’s suggestion, we now provide **a more direct comparison between DroidSpeak [1] and ICaRus in Table R2.1.**
>
> > Table R2.1. Comparisons between DroidSpeak and ICaRus. \alpha, N, and L represent recomputation ratio, the number of models, and total sequence length.
>
> | Aspect              | DroidSpeak                                                                                              | ICaRus (Ours)                                                                                          |
> |---------------------|--------------------------------------------------------------------------------------------------------|--------------------------------------------------------------------------------------------------------|
> | **Training**        | Models are trained independently **without considering KV-cache sharing** .                                | Models are trained **with considering KV-cache sharing**. |
> | **KV-cache sharing**  | Reuses KV caches only in **a subset of “non-sensitive” layers**.                                           | Reuses identical KV caches at **all layers**.                                                         ||
> | **Accuracy**        | Accuracy degrades as more layers reuse shared KV caches.                                               | Accuracy is preserved even when all layers share KV caches.                                           |
> | **Robustness** | Low robustness: **calibration-chosen “important” layers may not match those under real traffic,** leading to unpredictable quality drops in production. | High robustness: models are trained under full KV-cache sharing, so **KV caches can be fully shared at inference time just as during training**, without unexpected degradations in response quality. |
> | **KV memory**       | $\mathcal{O}((1+\alpha( N -1)) L)$                                                                              | $\mathcal{O}(L)$                                                                                      |
> | **\# Prefill Compute** | $\mathcal{O}((1+\alpha( N -1)) L^2)$                                                                          | $\mathcal{O}(L^2)$
>
> We would like to note that DroidSpeak has not released its code and its system design is relatively complex to implement, so our comparison is based on a careful analysis of the published paper rather than on re-running an official implementation as reported in Table R.2.1.

---

> ### Author Response · Authors · 2025-11-19
> **Response to Reviewer 1HaB**
>
> **A1.**
>
> ---
> + **DroidSpeak**
>
> We would like to emphasize that DroidSpeak and ICaRus approach KV-cache sharing in fundamentally different ways. **DroidSpeak** starts from the insightful observation that a fine-tuned model can sometimes reuse the KV cache of its base model at a subset of layers without significant accuracy loss. Concretely, **the task-specific models are first trained independently without considering KV-cache sharing.** DroidSpeak then uses a calibration dataset to identify “non-sensitive’’ layers whose KV caches can be shared; at inference time, it reuses the base model’s KV cache at those layers, while recomputing the KV cache for the remaining “sensitive’’ layers. As reported in the DroidSpeak paper, **increasing the number of shared layers inevitably causes accuracy degradation,** because more sensitive layers are forced to share KV caches. From a robustness standpoint, **DroidSpeak’s reliance on a calibration dataset** to select “non-sensitive” layers makes it vulnerable to distribution shift. The previously selected layers may no longer be appropriate, and the system **cannot reliably guarantee high generation quality in actual serving scenarios** where the online distribution can deviate from the profiling data.
>
> From a systems perspective, **DroidSpeak must still maintain separate KV caches for the non-shared layers of each model.** In a multi-model inference scenario with $N$ models and total sequence length $L$, the base model needs to keep its full KV cache, and each of the remaining $N−1$ models must store and recompute KV caches for a subset of layers.  The number of such recomputed layers is proportional to the recomputation ratio $\alpha$ reported in DroidSpeak.  As a result, the total KV-cache memory and prefill computation scale as **$\mathcal{O}((1+\alpha (N-1))L)$** and **$\mathcal{O}((1+\alpha (N-1))L^2)$**, respectively. Although DroidSpeak reports recomputation ratio $\alpha \approx 0.1\text{–}0.3$, which does reduce memory and recomputation compared to a naïve multi-model system, **its cost still grows with both the number of models N and the sequence length L**, and thus remains challenging in agentic AI scenarios with many specialized models and long, evolving contexts.
>
> ---
> + **ICaRus**
>
> By contrast, **ICaRus is designed from the training stage so that different task-specialized models share the KV cache by construction**: when fine-tuning, we freeze the logical encoder and adapt only the task-specific logical decoder. **This paradigm shift allows multiple models to share the KV cache at all layers**, while maintaining **high accuracy** and **robustness in actual serving scenarios**, because the models are explicitly trained under the same “fully shared KV cache’’ condition used at inference time.
>
> Furthermore, since ICaRus fully shares the KV cache across models, the total KV-cache memory scales as $\mathcal{O}(L)$, and no prefill recomputation is needed across different models for the same prompt, so the prefill cost remains $\mathcal{O}(L^2)$. This yields substantial savings compared to DroidSpeak, whose memory and computation still grow with the number of models $N$. **These advantages become increasingly significant in real-world agentic AI scenarios, where the number of agents and the complexity of their interaction patterns continue to grow explosively.**
>
> ---
> + **Comparisons between fine-tuning costs and selective recomputations**
>
> We would like to emphasize that **both DroidSpeak and ICaRus require fine-tuning cost.** In other words, the cost of training adapters is incurred by both approaches, and when using a small dataset together with parameter-efficient fine-tuning (PEFT), **this fine-tuning cost is relatively small and paid only once.** By contrast, as discussed above, **the cost of selective recomputation in DroidSpeak grows with the number of agents and the sequence length** in agentic AI scenarios, and this cost is incurred repeatedly at inference time in real production systems.
>
> ---
> We thank the reviewer for these insightful comments, which helped us clarify the relationship between DroidSpeak and ICaRus and strengthen the paper. We have incorporated these clarifications into Section 1 and Section 3.2 of the revised manuscript, and we plan to conduct a more extensive empirical comparison if an official implementation of DroidSpeak becomes available.

---

> ### Author Response · Authors · 2025-11-19
> **Response to Reviewer 1HaB**
>
> **Q2.** The evaluation only considers a round-robin routing scheme, which deliberately increases the number of models being used in a session. It would be useful to see the evaluation under prefix-aware or KV-aware routing. Do the gains still hold in terms of the latency and throughput if reuse across models is not required?
>
> **A2.** We sincerely thank the reviewer for the valuable feedback.
>
> ---
> + **About round-robin routing scheme**
>
> **We adopted the round-robin routing scheme because this style of multi-agent interaction appears in many existing systems.** For example, the multi-agent debate pattern in **AutoGen [2]** runs multiple solver agents plus an aggregator on the same task, allowing them to take turns responding and to refine or compete with each other. Similarly, **MetaGPT [3]** constructs a software-engineering workflow in which role-specific agents (planning, architecture design, system design, coding, etc.) are invoked sequentially for a single user request, all sharing an evolving context. To reflect the prevalence of such round-robin interactions in practical multi-agent systems, **prior multi-agent inference works such as DroidSpeak [1] and KVFlow [5] also evaluate their methods under this representative, widely used round-robin pattern.** Following this line of work, our round-robin multi-LoRA setting models workflows in which a fixed set of agents is invoked sequentially on a shared prefix.
>
> Nevertheless, we fully understand the reviewer’s concern that, in more dynamic and skewed agentic workloads, where only a subset of agents is active at any given time, the opportunities for cross-model KV reuse may decrease and the performance gains of ICaRus could be reduced. To address this concern, we conducted **an additional experiment** in which the controller **invokes agents at random with a skewed workload under ReAct workflow, so that on a typical turn only a subset of agents is active,** better reflecting such real-world scenarios. Specifically, unlike the round-robin invocation pattern in Section~\ref{sec: performance on multi model inference}, we construct a skewed workload in which one agent is invoked with probability 50% on each turn, while the remaining agents share the rest of the probability mass and are invoked in a random order rather than a fixed sequence. The experiments are conducted on the vLLM v0 architecture and the results are reported below.
>
> > [Figure R2.1 P95 latency and throughput of ICaRus under the ReAct workflow where the agent invocation pattern is random and skewed.](https://drive.google.com/file/d/1zaAEAAbP3zDlw6O_nb2Ya8yeFvsja-ZP/view?usp=sharing)
>
>  Figure R2.1 shows that ICaRus maintains low P95 latency and high throughput under dynamic and skewed agentic patterns. For example, with 2 models at 0.4 QPS, ICaRus achieves **15× lower P95 latency and 1.2× higher throughput** than the baseline, demonstrating that **the core advantage of ICaRus, enabling per-model prefix caching on top of cross-model KV sharing,** is preserved even under skewed and random agent invocation patterns. Furthermore, in the baseline, throughput quickly saturates beyond a certain QPS because rapid growth of the KV cache triggers frequent evictions and recomputations. In contrast, ICaRus allows multiple models to share a single KV cache pool, **keeping entries within the available GPU memory budget without eviction** so that throughput continues to increase with QPS without saturation. As a result, in the 8-model setting, ICaRus achieves up to **3.5× higher throughput** than the baseline under skewed and dynamic agent invocation patterns.
>
> ---
> + **About prefix-aware or KV-aware routing**
>
> We sincerely apologize for not describing our routing and caching assumptions more clearly, which may have caused confusion. In our evaluation, we assume a multi-LoRA setting in which **multiple task-specific LoRA adapters share the same base model on a single GPU.** Under this setup, both the baseline multi-LoRA system and ICaRus **already leverage the standard prefix/KV-aware mechanisms** of the serving stack: requests routed to the same LoRA module reuse the existing KV cache for identical prefixes whenever possible, thereby sharing KV-cache memory and avoiding redundant prefill recomputation within each model. Even under this per-model prefix/KV-aware setting, ICaRus achieves up to **11.1× lower P95 latency** and **3.8× higher throughput** than the baseline multi-LoRA system, as reported in the manuscript. In summary, our contribution **is orthogonal to this per-model KV-aware reuse,** as ICaRus further **enables cross-model KV sharing across different task-specialized models** on top of these existing mechanisms.

---

> ### Author Response · Authors · 2025-11-19
> **Response to Reviewer 1HaB**
>
> **A2.**
>
> ---
>
> + **About the scenarios that reuse across models is not required**
>
> We note that **ICaRus is primarily designed for workloads where multiple task-specialized models (or agents) share substantial contextual prefixes** over the course of a session. In the degenerate case where effectively only a single model is used and there is almost no cross-model prefix overlap, the opportunity for cross-model KV sharing largely disappears, and the additional benefits of ICaRus over a strong per-model baseline naturally become limited. **For this reason, the gains of ICaRus grow as the number of models increases:** for example, in our experiments with 2 models, ICaRus achieves 3.8× lower P95 latency and 1.4× higher throughput than the baseline, whereas with eight models the gains increase to 11.1× lower P95 latency and 3.8× higher throughput, as reported in the manuscript.
>
> However, we would like to emphasize that **the recent trend in both research and practice is moving from single-model systems toward multi-model, multi-agent architectures.** Several studies have shown that multi-agent systems can substantially **outperform single-agent systems [5, 6]**, and **performance improves as the number of agents increases [6, 7]**. Industrial efforts follow a similar direction: for example, **NVIDIA [7]** has highlighted multi-agent systems composed of multiple fine-tuned small language models as a key part of the future of agentic AI, and products such as **Microsoft’s Copilot Studio [8]** are beginning to adopt this paradigm in real-world applications. Yet in these multi-agent scenarios, the inability to reuse KV caches across models leads to significant memory and computation overheads during inference. We believe that **ICaRus offers a principled way to remove this KV-cache bottleneck in emerging multi-agent, multi-model serving workloads, thereby making agentic AI systems more scalable and practical in real deployments.**
>
> ---
> We thank the reviewer again for these thoughtful comments. We revised the manuscript to clarify our routing and caching assumptions in Appendix A2.2, including the new results under dynamic and skewed agent workloads in Appendix F.

---

> ### Author Response · Authors · 2025-11-19
> **Response to Reviewer 1HaB**
>
> **Q3.** In the round-robin evaluation setup, how many different models are used? I missed this detail and would encourage the authors to clarify.
>
> **A3.** We apologize for not describing our evaluation setup more clearly. We evaluate scenarios **with 2, 4, and 8 distinct models,** as shown in Fig. 8 of the manuscript. In the baseline, we use NNN different LoRA adapters (with no KV sharing across models), whereas in ICaRus we fine-tune N logical decoders on a shared logical encoder, enabling full cross-model KV sharing. We revised the manuscript to state this evaluation setup explicitly in Section 4.1.

---

> ### Author Response · Authors · 2025-11-19
> **Response to Reviewer 1HaB**
>
> **Q4.** In Figure 2, please clarify the insight as the training loss curves look almost identical and I am missing the takeaway.
>
> **A4.** Thank you for pointing this out. We apologize that the manuscript does not clearly explain the intended takeaway of Figure 2. The purpose of this figure is to show that **ICaRus, which enforces KV-cache sharing from the training stage, can be optimized just as well as conventional fine-tuning methods,** even though it only trains the logical decoder.
>
> ---
> + **Insight 1: Training only the logical decoder is sufficient for task-specific adaptation**
>
> In deep learning, adapting only a subset of parameters is a well-established strategy. For example, LoRA shows that **updating only a very small fraction of the parameters can achieve accuracy comparable to full fine-tuning [9, 10].** ICaRus follows a similar philosophy, but at **the level of the model’s logical structure**: we decompose a decoder-only Transformer into a logical encoder and a logical decoder, and **train only the logical decoder** using a parameter-efficient method such as LoRA.
>
> Figure 2 shows that, despite freezing the logical encoder and updating only the logical decoder, the training loss of ICaRus converges almost identically to that of the baseline that fine-tunes the full model (or a standard LoRA-tuned model). This indicates **that restricting learning to the logical decoder is sufficient to capture the task-specific knowledge needed for adaptation,** even when the logical encoder is shared across models. This observation is consistent with Tables 2–4 in the manuscript, where ICaRus matches or even slightly outperforms the baselines across various tasks and models.
>
> ---
> + **Insight 2: Logical-decoder-only training acts as a regularization**
>
> The ICaRus training scheme can also be viewed as imposing a strong regularization. By freezing the logical encoder and allowing only the logical decoder to adapt, we constrain all task-specific models to reuse a common sequence representation while expressing their differences only through the decoder. This acts as a form of implicit regularization: the model **cannot train by arbitrarily changing the shared encoder,** and must instead learn task-specific behaviors within a restricted parameter subset.
>
> Empirically, we observe that **this constraint does not hinder optimization** (as seen in the nearly identical loss curves in Figure 2), and in some tasks it even leads to better generalization than the baselines, as reflected in the accuracy results in Tables 2–4. This suggests that **logical-decoder-only training can provide a useful regularization effect, beyond simply updating a physical subset of parameters.**
>
> ---
> We revised the manuscript to make these two insights explicit in the Section 3.2, and we plan to investigate the regularization and representation-learning effects of ICaRus in greater depth as future work.

---

> ### Author Response · Authors · 2025-11-19
> **Response to Reviewer 1HaB**
>
> [1] Liu, Y. et al., “DroidSpeak: KV Cache Sharing for Cross-LLM Communication and Multi-LLM Serving,” arXiv, 2024.
>
> [2] Wu, Q. et al., “AutoGen: Enabling Next-Gen LLM Applications via Multi-Agent Conversation Framework,” COLM 2024.
>
> [3] Hong, S. et al., “MetaGPT: Meta Programming for a Multi-Agent Collaborative Framework,” ICLR 2024.
>
> [4] Pan, Z. et al., “KVFlow: Efficient Prefix Caching for Accelerating LLM-Based Multi-Agent Workflows,” NeurIPS 2025.
>
> [5] Shen, W. et al., “Small LLMs Are Weak Tool Learners: A Multi-LLM Agent,” EMNLP 2024.
>
> [6] Subramaniam, V. et al., “Multiagent Finetuning: Self Improvement with Diverse Reasoning Chains,” ICLR 2025.
>
> [7] Colmena, I. et al., “The Cost of Dynamic Reasoning: Demystifying AI Agents and Test-Time Scaling from an AI Infrastructure Perspective,” arXiv, 2025.
>
> [8] Spataro, J., “Introducing Microsoft 365 Copilot: Tuning, Multi-Agent Orchestration, and More from Microsoft Build 2025,” Microsoft 365 Blog, 2025.
>
> [9] Hu, E. J. et al., “LoRA: Low-Rank Adaptation of Large Language Models,” ICLR 2022.
>
> [10] Schulman, J., “LoRA Without Regret,” Thinking Machines Lab: Connectionism, 2025.

---

> ### Author Response · Authors · 2025-11-27
> **Gentle Reminder regarding our Rebuttal**
>
> Dear Reviewer 1HaB,
>
> We would like to once again sincerely thank you for your careful review and positive evaluation of our work. As the discussion period is drawing to a close, we would like to respectfully remind you that we have thoroughly revised the manuscript to reflect all of your insightful comments, and we summarize the main changes in the revised version below.
>
> + In our response A1, we provide a detailed comparison between DroidSpeak and ICaRus, showing that **ICaRus overcomes the selective recomputation and robustness limitations in DroidSpeak**. Furthermore, while both DroidSpeak and ICaRus require fine-tuning, ICaRus does not incur the cost of selective recomputations; these contributions have been incorporated **into Sections 1 and 3.2 of the revised manuscript.**
>
> + In our response A2, we experiment that ICaRus **yields high performance even in practical serving scenarios** beyond simple round-robin scheduling, clarify that our experiments **already account for prefix-aware routing**, and provide additional evidence that **the demand for reuse across models is increasing**, where ICaRus can bring substantial gains, in both academia and industry; these clarifications and new experimental results have been incorporated **into Appendix A2.2 and F of the revised manuscript**.
>
> + In our response A3, we **clarify the evaluation setup,** and these clarifications have been incorporated **into Section 4.1 of the revised version.**
>
> + In our response A4, we **provide an analysis of the capacity and regularization effects in ICaRus** based on the training loss curves in Figure 2, and this analysis has been incorporated **into Section 3.2 of the revised manuscript.**
>
>
> We are very grateful for your constructive feedback, which has significantly improved the quality of the paper, and we hope that the revised version and our responses have adequately addressed your concerns. We look forward to your further feedback.
>
> Best regards,
>
> The Authors

---

### Official Review · Reviewer_7YdR · 2025-11-09

**Soundness:** 3
**Presentation:** 4
**Contribution:** 3
**Rating:** 6
**Confidence:** 2

**Summary:**

The core idea of the work is to attempt to re-use the KV for the prefill phase, compared to the traditional LoRA based approach.The work instead finetunes a decoder style model to handle the sub tasks, using a light weight adapter.

**Strengths:**

1. I think this is an interesting twist on traditional LoRA by choosing to change just the decoding side.
2. provides a 11x decent speedup over the existing gpu cache system

**Weaknesses:**

1. I feel like the optimization provided by Icarus, while very impressive, feels a bit too incremental on existing LoRA systems.

2. From Appendix A2.2, the experiments tested the case where each agent has it's own LoRA adapter. Due to arrival patterns and batching, It is unclear the amount of LoRA used per GPU might not be true in practice.

3. The main alternatives the paper lists are either recompute or save. It is possible that swap is a valid strategy.

**Questions:**

1. was swap space turned on for the experiments? Can all the KV cache be evicted/loaded back instead of recomputation?

---

> ### Author Response · Authors · 2025-11-19
> **Response to Reviewer 7YdR**
>
> **Q1.** I feel like the optimization provided by Icarus, while very impressive, feels a bit too incremental on existing LoRA systems.
>
> **A1.** Thank you for this comment and for acknowledging the practical benefits of ICaRus. We would like to clarify that ICaRus is not an incremental optimization on top of existing LoRA systems, but rather introduces **a different architectural paradigm for multi-model inference.**
>
> ---
> + **ICaRus enables cross-model KV cache sharing, which conventional LoRA systems fundamentally cannot do.**
>
>  Existing multi-LoRA systems treat each LoRA-tuned model as an independent decoder-only Transformer: even if the models share the same base weights, each model maintains and updates its own KV cache, and prefix caching is restricted to within a single model. As a result, identical prompts routed to different task-specialized models must rebuild their KV caches independently, leading to cache explosion, frequent evictions, and redundant recomputation.
>
> In contrast, ICaRus **explicitly enforces that all task-specialized models share a single logical encoder,** so that an identical prefix deterministically produces an identical KV cache for all models. This architectural property **allows KV caches to be fully shared across models** and **enables cross-model prefix caching**, eliminating redundant prefill computation and preventing KV cache explosion in multi-model and agentic workflows. Empirically, this leads to up to **11.1× lower P95 latency** and **3.8× higher throughput** compared to a strong multi-LoRA baseline integrated in the same vLLM serving stack.
>
> ---
> + **The core idea of ICaRus is the encoder/decoder factorization of decoder-only Transformers, and is orthogonal to LoRA itself.**
>
>  ICaRus is based on the observation that a decoder-only Transformer can be conceptually **decomposed into a logical encoder**, which constructs the KV cache, and **a logical decoder**, which consumes the next token using KV cache. In ICaRus, we **freeze the logical encoder and fine-tune only the logical decoder,** so that multiple task-specialized decoders (math, coding, instruction, etc.) all share the same encoder and thus the same KV cache for a given prefix.
>
> In our paper, we instantiate the task-specific logical decoders with LoRA adapters because LoRA is a widely adopted, efficient, and high-quality PEFT method. However, **the ICaRus architecture itself does not rely on LoRA** in any essential way and **can be combined with other adaptation schemes,** such as DoRA [1], prompt-tuning [2] or full fine-tuning for training the logical decoder. **The novelty of ICaRus lies in this encoder/decoder factorization and the resulting identical-cache property across models, rather than in any particular choice of adapter mechanism.**
>
> ---
> We revised the paper to more clearly emphasize that (i) ICaRus is, to the best of our knowledge, the first architecture that enables multiple decoder-only Transformers to fully share KV caches across all layers in Section 1, and (ii) LoRA is used only as one convenient instantiation of the task-specific logical decoders, rather than being the source of the contribution in Section 3.2.

---

> ### Author Response · Authors · 2025-11-19
> **Response to Reviewer 7YdR**
>
> **Q2.** From Appendix A2.2, the experiments tested the case where each agent has it's own LoRA adapter. Due to arrival patterns and batching, It is unclear the amount of LoRA used per GPU might not be true in practice.
>
> **A2.** Thank you for raising this point. In our inference setup, we **assume that each agent is associated with a distinct LoRA adapter,** and that all of these agents can in principle be invoked within the same turn. For clarity, Appendix A2.2 evaluates a **balanced round-robin setting** in which these agents are called in turn, so that all LoRAs are used and may need to coexist on the same GPU.
>
> ---
> + **Round-Robin Pattern Examples in Real Systems**
>
> **We emphasize that this style of round-robin, multi-agent interaction appears in many existing systems.** For example, the multi-agent debate pattern in **AutoGen [3]** runs multiple solver agents plus an aggregator on the same task, allowing them to take turns responding and to refine or compete with each other. Similarly, **MetaGPT [4]** constructs a software-engineering workflow in which role-specific agents (planning, architecture design, system design, coding, etc.) are invoked sequentially for a single user request, all sharing an evolving context. To reflect the prevalence of such round-robin interactions in practical multi-agent systems, **prior multi-agent inference works such as DroidSpeak [5] and KVFlow [6] also evaluate their methods under this representative, widely used round-robin pattern.** Following this line of work, our round-robin multi-LoRA setting models workflows in which a fixed set of agents is invoked sequentially on a shared prefix.
>
> ---
> + **Additional Experimental: Random and Skewed Agent Invocation Patterns**
>
> That said, we fully agree with the reviewer’s insightful observation that, in many recent practical and dynamic agentic patterns, the effective number of LoRAs actively used on a GPU at any given time depends heavily on the workload: arrival patterns, batching, and routing policies can easily produce skewed usage where a few agents dominate the traffic while others are rarely invoked.
>
> To address this concern, we conducted **an additional experiment** in which the controller **invokes agents at random with a skewed workload under ReAct workflow, so that on a typical turn only a subset of agents is active,** better reflecting such real-world scenarios. Specifically, unlike the round-robin invocation pattern in the manuscripts, we construct a skewed workload in which one agent is invoked with probability 50% on each turn, while the remaining agents share the rest of the probability mass and are invoked in a random order rather than a fixed sequence. The experiments are conducted on the vLLM v0 architecture and the results are reported below.
>
> > [Figure R1.1. P95 latency and throughput of ICaRus under the ReAct workflow where the agent invocation pattern is random and skewed.](https://drive.google.com/file/d/1zaAEAAbP3zDlw6O_nb2Ya8yeFvsja-ZP/view?usp=sharing)
>
>  Figure R1.1 shows that ICaRus maintains low P95 latency and high throughput under dynamic and skewed agentic patterns. For example, with 2 models at 0.4 QPS, ICaRus achieves **15× lower P95 latency** and **1.2× higher throughput** than the baseline, demonstrating that **the core advantage of ICaRus, enabling per-model prefix caching on top of cross-model KV sharing,** is preserved even under skewed and random agent invocation patterns. Furthermore, in the baseline, throughput quickly saturates beyond a certain QPS because rapid growth of the KV cache triggers frequent evictions and recomputations. In contrast, ICaRus allows multiple models to share a single KV cache pool, **keeping entries within the available GPU memory budget without eviction** so that throughput continues to increase with QPS without saturation. As a result, in the 8-model setting, ICaRus achieves up to **3.5× higher throughput** than the baseline under skewed and dynamic agent invocation patterns.
>
> ---
> We thank the reviewer for this insightful suggestion, which led us to confirm that ICaRus enjoys even larger advantages over conventional systems under realistic dynamic and skewed agentic workloads. We incorporated these results into the Appendix F of the revised version of the paper.

---

> ### Author Response · Authors · 2025-11-19
> **Response to Reviewer 7YdR**
>
> **Q3.** The main alternatives the paper lists are either recompute or save. It is possible that swap is a valid strategy. was swap space turned on for the experiments? Can all the KV cache be evicted/loaded back instead of recomputation?
>
> **A3.**
>
> ---
> + **Swap Configuration in Our Experiments**
>
> Thank you for this valuable question. As the reviewer points out, **we used only the recompute-based configuration and did not enable swap.** This is because, in the **vLLM V1**–based serving stack we adopt, only recompute is supported for KV cache management; **the earlier swap-based mechanism has been deprecated [7]** due to its consistently worse performance compared to recompute on our workloads.
>
> ---
>  + **Experimental Results with Swap Enabled**
>
>  Following the reviewer’s suggestion, **we conducted additional experiments with swap enabled** (4GB swap space) using an earlier version of vLLM that supports this feature.  The experimental results are reported below.
>
> > [Figure R1.2 P95 latency and throughput of ICaRus under the ReAct workflow in LLaMA3.1-8B under swap-based kv cache management](https://drive.google.com/file/d/1x73ZCNm6s8CDIdRoPMIr9-6Cz3HohDlt/view?usp=sharing)
>
>  Figure R1.2 shows that  ICaRus continues to provide **lower P95 latency and higher throughput** even when the multi-model system uses swap for KV cache management. In particular, with 8 LoRA modules, ICaRus achieves up to **12.1× lower P95 latency** and **3.8× higher throughput** than the baseline. This is because ICaRus reduces the KV cache footprint itself, so that **even at higher QPS the GPU does not saturate and expensive swap operations are rarely triggered in the first place.**
>
> In summary, we emphasize that **recompute/swap strategies and ICaRus address orthogonal aspects of the problem.** Concretely, recompute or swap determine how to manage KV cache once GPU memory becomes full (e.g., whether to evict and reload from host storage or to recompute), whereas ICaRus fundamentally reduces KV pressure by enabling **cross-model KV sharing** across task-specialized models. By avoiding redundant KV construction across models, ICaRus **effectively delays or mitigates the point at which the KV cache saturates GPU memory,** thereby improving performance regardless of whether the underlying system chooses recompute or swap as its eviction policy. In principle, ICaRus could also **be combined with swap-based KV management.**
>
> ---
>  We thank the reviewer again for raising this point and for encouraging us to examine swap-based configurations as well. We have incorporated these additional results and clarifications into Appendix E of the revised manuscript.

---

> ### Author Response · Authors · 2025-11-19
> **Response to Reviewer 7YdR**
>
> [1] Liu et al., “DoRA: Weight-Decomposed Low-Rank Adaptation”, ICML 2024.
>
> [2] Lester et al., “The Power of Scale for Parameter-Efficient Prompt Tuning”, EMNLP 2021.
>
> [3] Wu et al., “AutoGen: Enabling Next-Gen LLM Applications via Multi-Agent Conversation”, ICLR 2024 LLMAgents Workshop.
>
> [4] Hong et al., “MetaGPT: Meta Programming for Multi-Agent Collaborative Framework”, ICLR 2024.
>
> [5] Liu et al., “DroidSpeak: KV Cache Sharing for Cross-LLM Communication and Multi-LLM Serving”, arXiv, 2024.
>
> [6] Pan, Z. et al., “KVFlow: Efficient Prefix Caching for Accelerating LLM-Based Multi-Agent Workflows,” NeurIPS 2025.
>
> [7] vLLM Team, “vLLM V1 User Guide”, vLLM Online Documentation, 2025.

---

> ### Author Response · Authors · 2025-11-27
> **Gentle Reminder regarding our Rebuttal**
>
> Dear Reviewer 7YdR,
>
> We would like to once again sincerely thank you for your careful review and positive evaluation of our work. As the discussion period is drawing to a close, we would like to respectfully remind you that we have thoroughly revised the manuscript to reflect all of your insightful comments, and we summarize the main changes in the revised version below.
>
> + In our response A1, we clarify that ICaRus **introduces the new concepts of a logical encoder** and logical decoder to enable KV cache sharing across different models, and that it **can be instantiated with other training methods** beyond LoRA, such as prompt tuning, DoRA, and full-parameter fine-tuning. These clarifications have been incorporated into **Section 3.2 of the revised manuscript.**
>
> + in our response A2, we **conducted additional experiments under dynamic arrival patterns and batching** and demonstrate that ICaRus **achieves low P95 latency and high throughput** even in practical serving scenarios. The corresponding results and analysis have been added to **the revised version in Appendix F.**
>
> + In our response A3, we **evaluated a scenario in which evicted KV cache entries are swapped rather than recomputed**. We find that the fundamental advantages of ICaRus remain unchanged in this setting, and it **continues to deliver low P95 latency and high throughput.** These findings have been incorporated into the **revised version in Appendix E.**
>
> We are very grateful for your constructive feedback, which has significantly improved the quality of the paper, and we hope that the revised version and our responses have adequately addressed your concerns. We look forward to your further feedback.
>
> Best regards,
>
> The Authors

---

### Author Response · Authors · 2025-11-19
**To All Reviewers: Key Contributions of ICaRus**

We thank the reviewers for the insightful and constructive feedback.
Before addressing the detailed comments, we briefly emphasize the core value of ICaRus.

---
+ **A Paradigm Shift to Full Cross-Model KV Cache Sharing via a Logical Encoder–Decoder Architecture**

To the best of our knowledge, no existing architecture allows multiple models to fully share their KV caches, which has made collaboration between task-specialized expert models difficult and computationally expensive. **ICaRus is the first to enable multiple experts to fully share context without costly recomputation** by introducing the simple but powerful idea of logically decomposing a decoder-only Transformer into a shared logical encoder and task-specific logical decoders.

---
+ **Bridging Train and Serve: Training-Time KV Sharing for Robust Real-World Serving**

ICaRus explicitly **accounts for cross-model KV cache sharing already at training time** by freezing the logical encoder. This training strategy **not only enables high generation quality answer** even when multiple models fully share KV caches, **but also provides robustness in real-world serving**, since there is **no discrepancy between training and inference.** In contrast, prior recomputation-based approaches introduce a train–serve mismatch in how KV caches are handled, which makes their behavior riskier to rely on in production systems.

---
+ **Toward Scalable Multi Model Systems Beyond the KV-Cache Bottleneck**

In both academia and industry, systems are rapidly evolving from single-model setups to architectures in which multiple models are orchestrated together for complex tasks [1,2,3,4]. Several works have shown that orchestrating more expert models can lead to stronger overall capability [4, 5]. However, as the number of expert agents grows, efficiently managing each model’s KV cache has remained a major scalability bottleneck for multi-model systems. ICaRus addresses this limitation **by allowing the KV caches of even hundreds of agents to be managed within a single shared pool,** enabling a **qualitatively new regime of multi-model systems in which large numbers of specialized expert models can collaborate** to solve complex tasks that were previously out of reach for single-model or conventional multi-model architectures.

---
[1] Belcak et al., “Small Language Models Are the Future of Agentic AI”, arXiv, 2025

[2] Spataro, “Introducing Microsoft 365 Copilot Tuning, Multi-Agent Orchestration, and More from Microsoft Build 2025”, Microsoft 365 Blog, 2025.

[3] Shen et al., “Small LLMs Are Weak Tool Learners: A Multi-LLM Agent”, EMNLP 2024.

[4] Subramaniam et al., “Multiagent Finetuning: Self Improvement with Diverse Reasoning Chains”, ICLR 2025.

[5] Kim et al., “The Cost of Dynamic Reasoning: Demystifying AI Agents and Test-Time Scaling from an AI Infrastructure Perspective”, arXiv, 2025.

---

### Author Response · Authors · 2025-11-30
**Summary of Reviews and Rebuttal for the New Area Chair**

Dear New Area Chair.

We deeply regret the circumstances surrounding the recent incident and recognize the additional burden this has placed on the new Area Chair. We are very grateful that the Area Chair has stepped in at this late stage to carefully evaluate our submission under these unusual conditions.

To assist the assessment and, at the same time, to highlight how our rebuttal addressed the main concerns, we **summarize the review–rebuttal process in the table below.** For clarity, we **boldfaced questions that were raised by multiple reviewers,** as these represent the core points of contention around ICaRus.

|Reviewer|Weakness \& Question|Answer|Score(orig)|Score(rev)|
|-|-|-|-|-|
|7YdR|**Too incremental on exsiting LoRA systems.**|ICaRus is an encoder/decoder factorization that enables cross-model KV cache sharing, independent of LoRA and compatible with other adaptation methods.|6|6 (no ans.)|
||**The inference setup may not reflect realistic dynamic arrival patterns.**|We add experiments with [random, skewed multi-agent workloads](https://drive.google.com/file/d/1zaAEAAbP3zDlw6O_nb2Ya8yeFvsja-ZP/view?usp=sharing) that better reflect practical dynamic patterns, and ICaRus still substantially outperforms conventional systems.| | |
||Was swap space turned on for the experiments?|Swap wasn’t used in the main setup; we add experiments with [swap-based KV management](https://drive.google.com/file/d/1x73ZCNm6s8CDIdRoPMIr9-6Cz3HohDlt/view?usp=drive_link), and ICaRus still clearly outperforms the baseline by reducing KV pressure.| | |
|1HaB|Comparison with DroidSpeak|We add a [direct comparison with DroidSpeak](https://drive.google.com/file/d/1qPNcdzT0ytVKTc8YOMtkBGuvM-bpgTnD/view?usp=sharing): ICaRus is explicitly trained for KV cache sharing, leading to higher accuracy and robustness, while fully shared KV caches greatly reduce memory usage and prefill recomputation.|4|4 (no ans.)|
||**The inference setup may not reflect realistic dynamic arrival patterns** without considering KV-aware routing.|We clarified that our setup is designed for KV-aware multi-agent scenarios; we add experiments [with random, skewed multi-agent workloads](https://drive.google.com/file/d/1zaAEAAbP3zDlw6O_nb2Ya8yeFvsja-ZP/view?usp=sharing) that better reflect practical dynamic patterns, and ICaRus still substantially outperforms conventional systems.| | |
||How many different models are used in the round-robin evaluation setup?|We clarify in Section 4.1 that our setup uses $N=2,4,8$ distinct models.| | |
||What is the insight from Figure 2 if the training loss curves look almost identical?|The nearly identical curves show that (1) training only the logical decoder is sufficient for task-specific adaptation under a shared encoder, and (2) this decoder-only training acts as a regularizer that preserves (and sometimes improves) generalization.| | |
|eTzX|Unclear accuracy comparison to workflows without any sharing or naive baselines.|We already compare against naive single-model baselines in Tables 2 and 4, and we updated these tables to explicitly mark single vs. multi-model and KV-sharing settings.|6|8|
||How robust is this method on larger models and more datasets?|We add additional experiments [training Qwen3-32B on the ToolAce dataset](https://drive.google.com/file/d/1WbHK_irNML9GKvqQsDxZW8qUx4jlz3yx/view?usp=sharing) and confirm that ICaRus trains stably and achieves comparable or better accuracy than the baseline, demonstrating its robustness.| | |
||**How are KV caches handled for tokens generated during decoding, can they be reused across agents?**|KV caches generated during decoding can also be reused in ICaRus without recomputation, because all models share the same logical encoder.| | |
|QNML|Is it common to use multiple fine-tuned models instead of a single model with different prompts?|We argue that this multi-model setting is increasingly common and support this by presenting multiple research and industrial examples, and that ICaRus provides a principled solution to the KV-cache bottleneck in such workloads.|6|8|
||**The inference setup may not reflect realistic dynamic arrival patterns.**|We add experiments with [random, skewed multi-agent workloads](https://drive.google.com/file/d/1zaAEAAbP3zDlw6O_nb2Ya8yeFvsja-ZP/view?usp=sharing) that better reflect practical dynamic patterns, and ICaRus still substantially outperforms conventional systems.| | |
||**How are KV caches handled for tokens generated during decoding, can they be reused across agents?**|KV caches generated during decoding can also be reused in ICaRus without recomputation, because all models share the same logical encoder.| | |
||**Is ICaRus coupled with LoRA finetuning?**|ICaRus is an encoder/decoder factorization that enables cross-model KV cache sharing, independent of LoRA and compatible with other adaptation methods.| | |

---

> ### Author Response · Authors · 2025-11-30
> **Summary of Reviews and Rebuttal for the New Area Chair**
>
> Through **additional experiments and qualitative analysis** conducted during the rebuttal period, we believe that **the contributions of ICaRus have been substantially clarified and strengthened.** In particular, reviewers **eTzX and QNML** increased their scores **from 6 to 8** after considering our new results and explanations. While reviewers **7YdR and 1HaB,** due to the abnormal termination of the rebuttal process, did not have the opportunity to revise their scores, we are confident that they would likely have updated their evaluations in a similar positive direction, as **their main concerns closely mirror those of eTzX and QNML, which were addressed to those reviewers’ satisfaction.**
>
> To the best of our knowledge, **ICaRus is the first architecture that enables multi models to fully share KV caches while preserving robustness**, by explicitly enforcing KV-cache sharing during training through **a novel factorization into a shared logical encoder and task-specific logical decoders.** We believe that this new treatment of cross-model KV sharing will play a central role in the emerging paradigm of multi agent systems, **substantially reducing serving costs** and **making collaborative agentic workflows far more scalable and practical in real-world deployments**.
>
> We would once again like to thank the Area Chair for the time and care devoted to evaluating our work under these challenging circumstances. We sincerely appreciate the thoughtful consideration given to ICaRus and hope that this summary is helpful for the final decision.
>
> Best regards,
>
> The Authors

---

### Meta-Review · Area_Chair_3w9g · 2026-01-07

**Summary:**

This paper proposes an arch for multiagent LLM inference with full KV-cache sharing across all layers. The core idea is to factorize transformer into a shared logical encoder that produces KV caches and ask specic logicial decoder.

Overall, primary cocnerns are addressed during rebuttal and incorporated into the revision, which is ready for publication in ICLR.

**Reviewer Concerns:**

commons ones

- difference between incremental vs. LoRA

- experiential validation, such as missing set up details (the number of models, etc.)

- robustness across tasks/datasets/models -- this is strengthed via additional data/task including qwen3-32b

Most of these are addressed. The outstanding one includes -- the lack of a direct end-to-end experimental baseline against DroidSpeak due to the absence of released code; also the applicability is structurally limited to settings where multiple specialists share a frozen encoder.

**Reviewer Scores:**

Reviewer 7YdR (original score 6): likely stay the same given the that the reviewer’s questions were mostly systems-evaluation and positioning clarifications

Reviewer 1HaB (original score 4): I guess this might increast but at most 6

Reviewer eTzX (original score 6, revised to 8): The reviewer explicitly indicated the rebuttal resolved concerns and requested that the added experiments be included; with full participation, the score would be 8 or 6.

Reviewer QNML (original score 6, revised to 8): similar, increase to 8

---

### Decision · Program_Chairs · 2026-01-26

Accept (Poster)